# A new moisture tagging capability in the Weather Research and Forecasting Model: formulation, validation and application to the 2014 Great Lake-effect snowstorm

Damián Insua-Costa[1] and Gonzalo Miguez-Macho[1]

[1]Non-Linear Physics Group, Universidade de Santiago de Compostela, Galicia, Spain

*Correspondence to:* Damián Insua Costa (damian.insua@usc.es) and Gonzalo Miguez Macho (gonzalo.miguez@usc.es)

**Abstract.** A new moisture-tagging tool, usually known as water vapor tracer (WVT) method or online Eulerian method, has been implemented into the Weather Research and Forecasting (WRF) regional meteorological model, enabling it for precise studies on atmospheric moisture sources and pathways. We present here the method and its formulation, along with details of the implementation into WRF. We perform an in-depth validation with a one month long simulations over North America at 20km resolution, tagging all possible moisture sources: lateral boundaries, continental, maritime or lake surfaces and initial atmospheric conditions. We estimate errors as the moisture or precipitation amounts that cannot be traced back to any source. Validation results indicate that the method exhibits high precision, with errors considerably lower than 1% during the entire simulation period, for both precipitation and total precipitable water. We apply the method to the Great Lake-effect snowstorm of November 2014, aiming at quantifying the contribution of lake evaporation to the large snow accumulations observed in the event. We perform simulations in a nested domain at 5km resolution with the tagging technique, demonstrating that about 30-50% of precipitation in the regions immediately downwind, originated from evaporated moisture in the Great Lakes. This contribution increases to between 50-60% of the snow water equivalent in the most severely affected areas, which suggests that evaporative fluxes from the lakes have a fundamental role in producing the most extreme accumulations in these episodes, resulting in the highest socio-economic impacts.

## 1 Introduction

Water is the most important natural resource in the planet, and without its presence, no form of life would be possible. Small changes in Earth's water transport and redistribution, as well as in sources and sinks of atmospheric moisture, can therefore result in enormous socio-economic impacts (Oki and Kanae, 2006). Detailed knowledge of the hydrologic cycle and its potential future alterations is thus of great relevance, and in particular of extreme hydrometeorological events, such as droughts and high precipitation episodes, which can cause catastrophic consequences in the very short-term (Easterling et al., 2000). Among the many uncertainties around the water cycle, researchers have tried to respond two questions with special interest: what are the moisture source regions for precipitation?, and, what are the consequences for precipitation of possible future changes in source regions due to anthropogenic influences or natural variability? To answer these fundamental questions, different numer-

ical methods have been applied in the last decades, namely analytical, Lagrangian and Eulerian models (e.g. Gimeno et al., 2012, for a detailed review).

Analytical models derived from the conservation equation of atmospheric water mass (Peixoto and Oort, 1992) have been widely used in calculations of the recycling ratio, which quantifies the contribution of local evapotranspiration to precipitation (Brubaker et al., 1993; Eltahir and Bras, 1996; Trenberth, 1999; Rios-Entenza and Miguez-Macho, 2014). A great advantage of these methods is their simplicity and low computational cost, at the expense, however, of strong assumptions, such as that water vapor of all origins is well mixed in the column (Budyko, 1974), that limit their applicability. For this reason, analytical models can only provide a first order estimation of the recycling ratio. In more recent years, these models have been refined, and some of the former initial assumptions have been relaxed. Some newer analytical models can quantify the contribution of remote moisture sources to local precipitation, while improving recycling ratio calculations (Dominguez et al., 2006). Nevertheless, most models still assume that moisture of all origins is well-mixed in the atmospheric column, notwithstanding some attempts to relax the hypothesis (Burde et al., 2006), and this can significantly compromise their results (Bosilovich, 2002).

Offline Eulerian methods, the so-called 2-D moisture tracking models (Yoshimura et al., 2004; Van Der Ent et al., 2010; Goessling and Reick, 2011), are an alternative to traditional analytical models especially useful for calculations of continental moisture recycling ratios on a global scale. This method uses vertically integrated variables for calculations, and hence still assumes the well-mixed atmosphere hypothesis, which leads to errors particularly in regions of significant vertical shear (Goessling and Reick, 2013). However, in recent years this hypothesis has been relaxed by adding an additional vertical level to some offline Eulerian models (i.e., moving from a single column to two layers), which has considerably improved the results provided by this method (Van Der Ent et al., 2013).

Lagrangian models, based on the spatio-temporal tracking of individual fluid particles, are possibly the most extended method to study sources and sinks of moisture. There are currently two main classes of Lagrangian models: the method of quasi-isentropic back-trajectories (Dirmeyer and Brubaker, 1999) and the method of dispersion of Lagrangian particles (Stohl et al., 2004). Lagrangian models have been extensively used in climatic studies of atmospheric water vapor sources (Stohl and James, 2005; Gimeno et al., 2010) and in the diagnosis of the origin of moisture in extreme precipitation events (James et al., 2004; Stohl et al., 2008). Among the advantages of the method are computational efficiency, that source regions to analyze do not need to be selected a priori, since particles can be traced back in time, and furthermore, that when using reanalysis data for calculations, they effectively introduce an observational constraint. Lagrangian models include, nevertheless, some simplifications in their formulation that can result in serious biases. For example, the method of dispersion of Lagrangian particles does not allow for a clear separation between evaporation (E) and precipitation (P), in addition to neglecting liquid water and ice, which results in an overestimation of both E and P. For its part, the method of quasi-isentropic back-trajectories does not have this limitation, since evaporation and precipitable water content are needed for calculations; however, the well-mixed atmosphere hypotheses is still invoked, since water from surface evaporation is assumed to contribute uniformly throughout the column; and moreover, phase changes along the path of the parcels are not considered. Apart from errors derived from approximations in the specific formulation of the hydrological part of each method, a common drawback to all Lagrangian models is the growing uncertainty in air parcel trajectories with time (Stohl, 1998). An important reason for this error comes

from the existence of sub-grid vertical motions, related to convection and turbulent transport, which are not resolved by the gridded atmospheric analyses that Lagrangian models use for calculations. Estimations of the effect of these processes must be made; however, the mere existence of subgrid vertical mixing in the column inevitably leads to imprecision in determining parcel trajectories , which is especially critical when studying variations in the moisture content of the parcel, since atmospheric

mixing ratios can change abruptly with height. Some of the aforementioned limitations of the Lagrangian method could be avoided considerably by using the output of a climate model (e.g. Brioude et al., 2013), thereby obtaining a more detailed information about the meteorological variables needed for an improved particle tracking. Notwithstanding, with this strategy the observational constraint disappears and the computational cost increases substantially, effectively offsetting the main benefits of the Lagrangian method.

Online Eulerian methods, generally known as water vapor tracers (WVTs) are based on coupling a moisture tagging technique with a global or regional meteorological model. This strategy enables WVTs to fully consider all physical processes affecting atmospheric moisture, such as advection, molecular and turbulent diffusion, convection and cloud microphysics, thereby avoiding errors associated with offline methods. For this reason, this is presently regarded as the most accurate technique for the study of atmospheric moisture sources for precipitation. It has, nevertheless, some shortcomings; mainly related

to the fact that it implies running an atmospheric model and relying on results from the simulation, since the method cannot be applied a posteriori, i.e. based for example on atmospheric analyses. Biases in WVTs are therefore not so much linked to the strategy itself, but to the model where they are coupled; hence the method provides sound results only if the atmospheric model simulation is realistic. In addition, the associated computational cost is much higher than in any of the other techniques mentioned above.

WVTs were introduced in general circulation models in the early studies of Koster et al. (1986) and Joussaume et al. (1986). There were successive later implementations in different global models (Numaguti, 1999; Werner et al., 2001; Bosilovich and Schubert, 2002; Noone and Simmonds, 2002; Bosilovich et al., 2003; Goessling and Reick, 2013; Singh et al., 2016), all of them proving very useful in climatic studies of precipitation moisture sources. WVTs in global models allow for investigations at the planetary scale, covering all existing moisture source regions. However, given the coarse resolution common to most

of these models, some processes such as surface hydrology or water vapor transport in complex topography areas, are subject to sizeable biases, which compromise conclusions drawn from the WVTs method. WVTs in regional climate models, which employ a much finer resolution and significantly improve the representation of small-scale features of the hydrology cycle, are perhaps the best alternative for diagnosing precipitation moisture sources in events of reduced temporal and spatial scale, such as extreme precipitation episodes. They can also be useful in climatic studies at the regional scale. The first implementation of

the moisture tagging capability in a regional atmospheric model was in the CHRM model Sodemann et al. (2009), and more have followed since in different models (Knoche and Kunstmann, 2013; Miguez-Macho et al., 2013; Winschall et al., 2014; Arnault et al., 2016).

Although the different implementations of WVTs in global or regional models have in common the theoretical approach, they can, nevertheless, be somewhat different in practice. These differences are not only due to the model or parameterizations

used, but also to the considerations and simplifications that authors assume in their own implementations, which can potentially

**Table 1.** The different WVTs implementations (including the present): reference, name of the models in which the WVTs tool has been implemented and scale of these models.

| Reference of the implementation | Model name | Model scale |
|---|---|---|
| Joussaume et al. (1986) | LMD | Global |
| Koster et al. (1986) | NASA/GISS | Global |
| Numaguti (1999) | CCSR-NIES | Global |
| Werner et al. (2001) | ECHAM4 | Global |
| Bosilovich and Schubert (2002) | GEOS-3 | Global |
| Noone and Simmonds (2002) | MUGCM | Global |
| Bosilovich et al. (2003) | FVGCM | Global |
| Sodemann et al. (2009) | CHRM | Regional |
| Goessling and Reick (2013) | ECHAM6 | Global |
| Knoche and Kunstmann (2013) | MM5 | Regional |
| Miguez-Macho et al. (2013) | WRF 3.4.1 | Regional |
| Winschall et al. (2014) | COSMO | Regional |
| Arnault et al. (2016) | WRF 3.5.1 | Regional |
| Singh et al. (2016) | CAM5 | Global |
| Insua-Costa and Miguez-Macho (2018) | WRF 3.8.1 | Regional |

lead to significant inaccuracies. It is therefore fundamental to validate the method's precision before it can be reliably applied in practical cases.

This paper presents a new moisture tagging tool recently added to the Weather Research and Forecasting (WRF V3.8.1) regional meteorological model (WRF-WVT hereafter). Even though a preliminary version of the tool has already been tested in an older version of the model (Miguez-Macho et al., 2013; Dominguez et al., 2016; Eiras-Barca et al., 2017), we discuss here the formulation and implementation details of the method, and perform a thorough validation, thus avoiding the reliability uncertainty of which many other implementations of the kind suffer. The study is structured as follows: Section 2 describes the formulation and implementation into WRF of the WVTs method. Section 3 contains the validation strategy and results. Section 4 shows results from an example application and finally Section 5 includes a summary and conclusions of the work.

## 2 Implementation of the moisture tagging capability

### 2.1 General formulation

The basis of the moisture tagging technique is to replicate for moisture tracers the prognostic equation for total moisture:

$$\frac{\partial q_n}{\partial t} = -\boldsymbol{v} \cdot \nabla q_n + \nu_q \cdot \nabla^2 q_n + \left(\frac{\partial q_n}{\partial t}\right)_{PBL} + \left(\frac{\partial q_n}{\partial t}\right)_{microphysics} + \left(\frac{\partial q_n}{\partial t}\right)_{convection} \tag{1}$$

where $q_n$ refers to the different moisture species considered, namely water vapor, cloud water, rain water, snow, ice and graupel. The first two terms on the right hand side in Eq. (1) represent the tendencies due to advection and molecular diffusion, respectively, and the others correspond to tendencies resulting from parameterized turbulent transport (Planetary Boundary Layer, PBL scheme), microphysics and convection. The latter three terms account for subgrid physical processes affecting atmospheric moisture, such as phase changes and precipitation, or redistribution by convection and turbulent diffusion.

To replicate Eq. (1), six new variables $tq_n$ are created corresponding to the tracers of the different moisture species: tvapor, tcloud, train, tsnow, tice, tgraupel. We note that in earlier studies, only water vapor was tagged (tvapor), hence the name of Water Vapor Tracers (WVTs) method. Perhaps this denomination is no longer accurate when tagging all six moisture species, and more properly the technique should be referred to as simply Moisture Tracers (MTs) method; we will keep in the text, however, the common WVTs term, as it is already well established in the literature.

The general form of the prognostic equation for WVTs is totally analogous to Eq. (1), just replacing $q_n$ by $tq_n$. The Eulerian form of this equation and the fact that it is solved simultaneously with Eq. (1), are the reasons for the method to be classified as "online" Eulerian. One could think that since the prognostic equations for WVTs and total moisture have the same form, it would suffice with repeating the calculations performed for total moisture species for the tracer species, and just change initial or boundary conditions. However, this is not the case, since the behavior of the tagged moisture is not independent from that of

total moisture. In other words, the tagged moisture does not evolve as if it was completely on its own. A very simple example of this is saturation conditions and phase changes, which would hardly occur if only tagged moisture were considered. When an air parcel saturates, it does so in regards to its total moisture content, independently of whether its tracer moisture content is high or low. Similarly, since it is total moisture that determines the thermodynamical setting for turbulence and convection, primary and derived variables in the basis of the parameterizations of those processes, such as virtual temperature, dew point,

profile instability, convective available potential energy (CAPE), level of free convection, eddy diffusivity, and many more, must be computed using total moisture, even when calculations are performed for tagged moisture tendencies. Therefore, the prognostic equations for tracer moisture must be solved coupled to the governing equations of the model, i.e. "online", although tracer variables do not appear elsewhere and hence do not have an effect in the model's dynamics in any way.

    Thus, for the implementation of WVTs into WRF, three fundamental parameterizations of the model such as the turbulence

(PBL) scheme, microphysics and convection, must be modified for calculating the associated tracer moisture tendencies, as discussed above. Conversely, advection and diffusion routines can be simply called for tracers in the same way as for total moisture or any other scalar, since in these processes tracer moisture can indeed be treated independently from total moisture.

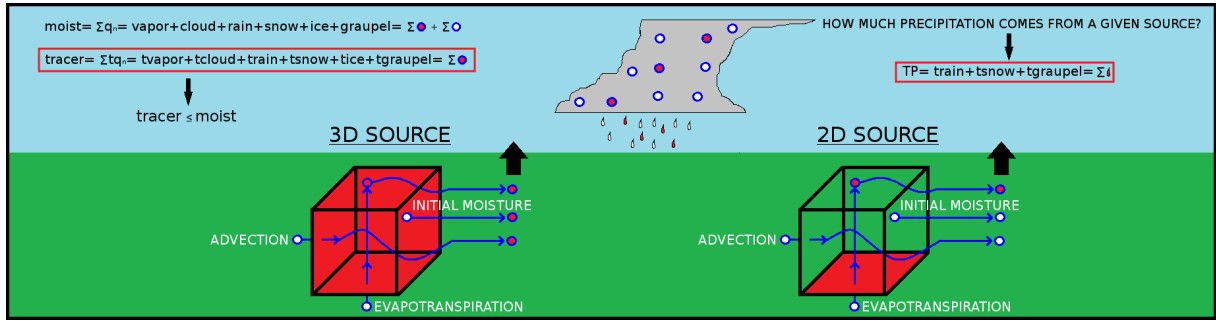

**Figure 1.** Sketch representing the fundamentals of the moisture tracers method, including the tagging of 3D and 2D moisture sources.

We note that it is important to use an advection numerical scheme that is positive definite, conserves mass and minimizes numerical diffusion, in order to limit numerical errors in WVT calculations. Both total moisture and tagged moisture must use the same scheme. All other components of the model remain unchanged, since they do not affect moisture dynamics directly.

## 2.2 Moisture tracer tendencies formulation

Of the several scheme options available, we have altered for moisture tagging the Yonsei University (YSU; Hong et al., 2006) PBL scheme, the WRF-Single-Moment 6-class (WSM6; Hong and Lim, 2006) microphysics scheme and the (Kain, 2004) convective parameterization. These schemes have been selected because they are some of the most commonly used and show a reliable performance in numerous situations.

### 2.2.1 Boundary layer parametrization

The equation of turbulent diffusion for moisture (Hong et al., 2006):

$$\left(\frac{\partial q_n}{\partial t}\right)_{PBL} = \frac{\partial}{\partial z}\left[K_q\left(\frac{\partial q_n}{\partial z} - \gamma_q\right) - \overline{(w'q'_n)}_h\left(\frac{z}{h}\right)^3\right] \tag{2}$$

is solved in this parameterization for $q_n$ =water vapor, cloud water and ice, with boundary conditions:

$$\begin{cases} \eta = 1 \implies K_q\left(\frac{\partial q_n}{\partial z}\right) = -\frac{Q_E}{\rho_{air}} \\ \eta = \eta_{end} \implies K_q\left(\frac{\partial q_n}{\partial z}\right) = 0 \end{cases} \tag{3}$$

where $Q_E$ represents the water vapor flux at the surface.

To compute turbulent diffusion for tracer species, we replicate Eq. (2), keeping the same eddy diffusivity coefficients $K_q$, turbulent vertical velocity $w'$ and boundary layer height $h$ as in the total moisture calculation:

$$\left(\frac{\partial tq_n}{\partial t}\right)_{PBL} = \frac{\partial}{\partial z}\left[K_q\left(\frac{\partial tq_n}{\partial z} - \gamma_{tq}\right) - \overline{(w'tq'_n)}_h\left(\frac{z}{h}\right)^3\right] \tag{4}$$

Boundary conditions are analogous to Eq. (3):

$$
\begin{cases}
\eta = 1 \implies K_q \left( \frac{\partial tq_n}{\partial z} \right) = -\frac{TQ_E}{\rho_{air}} \\
\eta = \eta_{end} \implies K_q \left( \frac{\partial tq_n}{\partial z} \right) = 0
\end{cases}
\tag{5}
$$

considering that now $TQ_E$ is the tracer water vapor flux at the surface, which, when upward, is equal to that of total water vapor in the areas that are selected for tagging, and zero in the rest.

### 2.2.2 Microphysics parameterization

The tendencies computed in the WSM6 microphysics parameterization account for grid scale precipitation and for the different phase changes among the several species considered (water vapor, cloud water, rain water, ice, snow and graupel):

$$
\left( \frac{\partial q_n}{\partial t} \right)_{microphysics} = \sum_x \frac{\partial Q_{q_x \to q_n}}{\partial t} - \sum_x \frac{\partial Q_{q_n \to q_x}}{\partial t} - \frac{q_n}{\rho_{air}} \frac{\partial}{\partial z} (\rho_{air} \cdot V_{q_n})
\tag{6}
$$

where $Q_{q_x \to q_n}$ and $Q_{q_n \to q_x}$ refer to the amount of moisture species $q_x$ transformed via phase change into moisture species $q_n$ and viceversa, respectively (see Hong and Lim, 2006, for details). The last term on the right hand side of Eq. (6) represents the tendency due to hydrometeor $q_n$ fallout, with an associated mass-weighted mean terminal velocity $V_{q_n}$. In the latter case, $q_n =$ refers only to rain water, snow, ice or graupel.

We consider that phase changes among the different tracer species occur in amounts proportional to their total moisture counterparts:

$$
TQ_{tq_x \to tq_n} = \frac{tq_x}{q_x} \cdot Q_{q_x \to q_n} \qquad\qquad TQ_{tq_n \to tq_x} = \frac{tq_n}{q_n} \cdot Q_{q_n \to q_x}
\tag{7}
$$

where the proportionality coefficients in Eq. (7) correspond to the tracer fraction in the species undergoing the change ($tq_x/q_x$ when $tq_x$ changes phase, and $tq_n/q_n$ when $tq_n$ does).

Bearing the latter consideration in mind, we replicate Eq. (6) to calculate moisture tracers' tendencies:

$$
\left( \frac{\partial tq_n}{\partial t} \right)_{microphysics} = \sum_x \frac{\partial TQ_{tq_x \to tq_n}}{\partial t} - \sum_x \frac{\partial TQ_{tq_n \to tq_x}}{\partial t} - \frac{tq_n}{\rho_{air}} \frac{\partial}{\partial z} (\rho_{air} \cdot V_{q_n})
\tag{8}
$$

Sedimentation processes yielding precipitation rates are computed in this WSM6 parameterization with a forward semi-Langrangian advection scheme with mass conservation and positive definition (Henry Juang and Hong, 2010), from which total accumulated grid-scale rain, snow and graupel are obtained. Applying the same strategy, we obtain the corresponding precipitation amounts for tracers. The ratio of tracer rain, snow and graupel to their total counterparts provides information about the contribution of the selected moisture sources to precipitation.

### 2.2.3 Convective parameterization

Following the formalism in Bechtold et al. (2001), the effect of convection in moisture can be generally described as:

$$\left(\frac{\partial q_n}{\partial t}\right)_{convection} = \frac{1}{\rho_{air} \cdot A} \left[\frac{\partial}{\partial z}(M^u + M^d)q_n - (\epsilon^u + \epsilon^d)q_n + \delta^u q_n^u + \delta^d q_n^d\right] + S_{q_n} \tag{9}$$

where $A$ is the grid cell area, $M^u$ and $M^d$ are the mass fluxes in updraft and downdraft, $\epsilon^u - \epsilon^d$ and $\delta^u - \delta^d$ represent mass exchanges between cloud and environment in the updraft and downdraft, due to entrainment and detrainment processes, respectively; $q_n^u$ and $q_n^d$ refer to the moisture amounts present in updraft and downdraft, and finally $S_{q_n}$ corresponds to sources and sinks of moisture species $q_n$ in the convective cloud, linked to phase changes and precipitation. The Kain-Fritsch parameterization considers up to five moisture species $q_n$ =water vapor, cloud water, rain water, snow and ice, but not all are equally treated, and simplified forms of Eq. (9) are used for some of them (see Kain and Fritsch, 1990; Kain et al., 2003; Kain, 2004, for further details).

Similarly to the previously discussed parameterizations, we replicate the general equation for convective moisture tendencies (Eq. (9)) for the case of tracers:

$$\left(\frac{\partial tq_n}{\partial t}\right)_{convection} = \frac{1}{\rho_{air} \cdot A} \left[\frac{\partial}{\partial z}(M^u + M^d)tq_n - (\epsilon^u + \epsilon^d)tq_n + \delta^u tq_n^u + \delta^d tq_n^d\right] + S_{tq_n} \tag{10}$$

where the proportionality assumption of Eq. (7) is applied again to calculate amounts in tracer phase changes.

In the Kain-Fritsch parameterization, a large fraction of the liquid water or ice that forms in the updraft is converted to precipitation (Kain et al., 2003), which can evaporate or sublimate on the way to the ground, resulting finally in total accumulated cumulus precipitation. The replication of these processes for tracers yields cumulus tracer precipitation. As in the case of the microphysics scheme, the ratio of tracer to total precipitation quantifies the existing contribution from the selected moisture sources.

### 2.2.4 Tracers initialization and boundary conditions

Tracer initial and lateral boundary conditions are usually set to zero, even though this does not always have to be the case, as we show when we perform the validation of the method in Section 3 and in the nested simulation discussed in Section 4. Lower boundary conditions depend largely on the moisture source to analyze. The implementation that we present here of the WVTs method, allows for the tracking of moisture from two and three-dimensional sources.

**a. 2D source**

Working with a two dimensional source commonly refers to tagging surface evapotranspiration fluxes ($Q_E$) from a certain region or interest $A_{2D}$. The flux of tracer water vapor at the surface $TQ_E$ can we written as:

$$\begin{cases} Q_E(x,y,t) > 0 \implies TQ_E(x,y,t) = Q_E(x,y,t) \quad \forall(x,y,t) \in A_{2D} \\ Q_E(x,y,t) < 0 \implies TQ_E(x,y,t) = \frac{tvapor(x,y,\eta=1,t)}{vapor(x,y,\eta=1,t)} \cdot Q_E(x,y,t) \quad \forall(x,y,t) \end{cases} \tag{11}$$

Negative fluxes indicate dew (or frost) deposition, and in this case, we use again the proportionality assumption for phase changes, as elsewhere in the atmosphere (Eq. (7)). The tracer deposition flux is simply the total deposition flux times the tracer fraction in the water vapor of the first atmospheric level. The resulting flux $TQ_E$ is used in Eq. (5) as lower boundary condition for moisture turbulent diffusion in the PBL parameterization.

**b. 3D source**

Any three dimensional volume $V_{3D}$ can be set as a 3D source for moisture tagging. This can refer to the entire atmosphere over a region of interest, or to only a part of it (for example the stratosphere). Setting the lateral boundaries plus the adjacent relaxation zone as 3D wall-like source regions is also the most convenient strategy for tagging incoming moisture fluxes from the exterior of the regional model domain.

To turn any given set of model domain points $V_{3D}$ into a 3D source for moisture tracers, we simply impose:

$$tq_n(x,y,\eta,t) = q_n(x,y,\eta,t) \quad \forall(x,y,\eta,t) \in V_{3D} \tag{12}$$

## 3    Moisture tracers validation

### 3.1    Experimental setup

The validation simulation for the newly implemented moisture tagging tool is performed with the WRF model version 3.8.1
(Skamarock et al., 2008), for the duration of one month (November 2014) and with a domain D1 of 20km horizontal resolution and 35 vertical levels (Fig. 2). Initial and boundary conditions, updated every six hours, were obtained from the National Centers for Environmental Prediction (NCEP) Final (FNL) Operational Model Global Tropospheric Analyses, available at $1^o$ resolution (National Centers for Environmental Prediction, National Weather Service, NOAA, 2000). In addition to the YSU PBL, WSM6 microphysics and Kain-Fritsch convective parameterizations that we adapted to calculate the corresponding tracer
tendencies (as described in Section 2), in the simulations, we also use the Noah Land Surface Model (Noah LSM; Chen and Dudhia, 2001) and the Rapid Radiative Transfer Model (RRTM; Mlawer et al., 1997) and Dudhia (Dudhia, 1989) schemes for long and shortwave radiation, respectively. Moisture and tracer advection are calculated with the 5th order Weighted Essentially Non-Oscillatory (WENO; Liu et al., 1994) scheme with positive definite limiter. Spectral nudging of waves longer than around 1000 km is activated to avoid distortion of the large scale circulation within the regional model domain due to the interaction
between the model's solution and the lateral boundary conditions(Miguez-Macho et al., 2004).

### 3.2    Methodology

The methodology followed to validate WVTs is analogous to that used previously by Bosilovich and Schubert (2002) and Sodemann et al. (2009), and it is based on tagging moisture from all possible sources, so that if the method were exact, the difference between tracer and total moisture should be zero. In other words, let $S_n$ (with $n = 1, 2, 3 \dots$) be a set of moisture

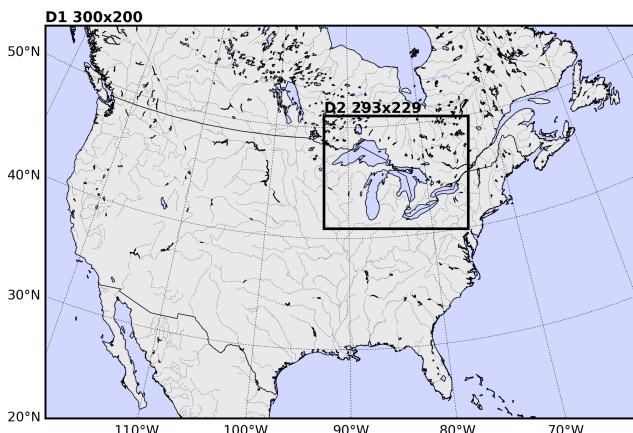

**Figure 2.** Simulation domains for the validation (D1) and example application experiments (D2).

sources covering all possible atmospheric moisture sources, $tq_{S_n}$ the total moisture (the sum of all moisture species) from each source $S_n$, and $q$ the total moisture, then the absolute error ($\varepsilon_a$) of the method can be written as:

$$\varepsilon_{a_{tq}}(x,y,\eta,t) = \sum_n tq_{S_n}(x,y,\eta,t) - q(x,y,\eta,t) \tag{13}$$

or in terms of precipitable water, integrating Eq. (13) in the vertical yields:

$$\varepsilon_{a_{TTPW}}(x,y,t) = \sum_n TTPW_{S_n}(x,y,t) - TPW(x,y,t) \tag{14}$$

where $TTPW_{Sn}$ refers to the total precipitable water coming from source $S_n$ and $TPW$ is the total precipitable water simulated by the model. Similarly, for precipitation:

$$\varepsilon_{a_{TP}}(x,y,t) = \sum_n TP_{S_n}(x,y,t) - P(x,y,t) \tag{15}$$

where $TP_{S_n}$ corresponds to the precipitation from source $S_n$ and $P$ is the total precipitation produced by the model. Equation (15) can also be applied to any particular type of precipitation, such as rain, snow or graupel, individually.

Here, we have divided the possible moisture sources into five ($S_1, \ldots, S_5$), three of them two-dimensional (Fig. 3a) and two three-dimensional (Fig. 3b). The two-dimensional source regions cover all evaporative sources within the domain, namely sea, land and lakes, whereas the three-dimensional sources tag incoming moisture from the lateral boundaries and the moisture contained in the full atmospheric volume of the domain at initial time. For the latter purpose, the three-dimensional source "INITIAL" (Fig. 3b) is activated only at the first time step of the simulation. The "BOUNDARY" source (Fig. 3b) is a wall-like volume encompassing the relaxation zone where lateral boundary conditions are applied, along the domain's outer edges. To

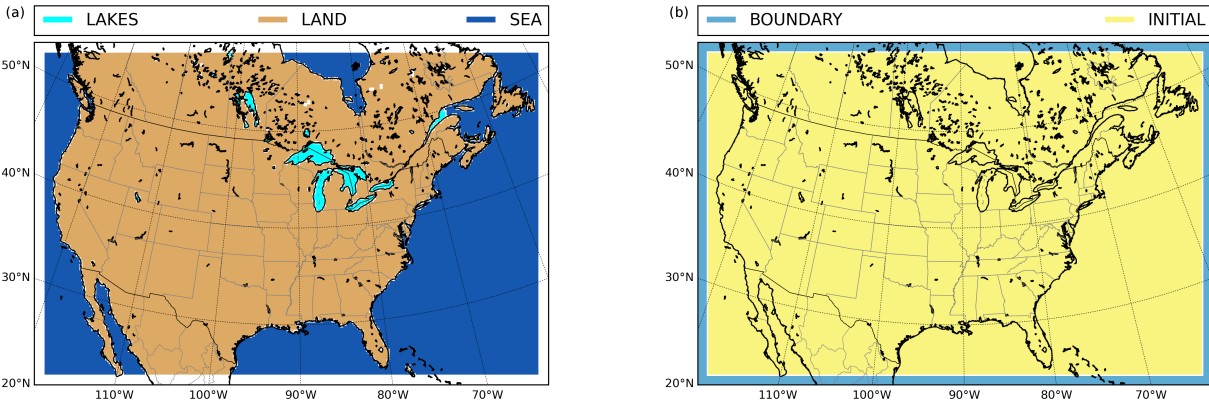

**Figure 3.** Moisture sources considered for validation calculations: two-dimensional (a) and three-dimensional (b).

prevent moisture from evaporative or initial condition sources to be counted twice, this boundary volume becomes a sink for tracers of these other origins, that is, tagged moisture species from other sources are set to zero when they enter "BOUNDARY". All possible atmospheric moisture sources are covered by the aforementioned five, and therefore Eq. (13), (14) and (15) should be fulfilled at all times with zero error, should the method were perfectly accurate.

To provide insights on the temporal evolution of the error, we follow the statistical treatment of Bosilovich and Schubert (2002), based on the calculation of the mean (ME) and standard deviation (SD) of the error at each point in time, that can be written as (following the notation used previously):

$$
\begin{cases}
ME = \frac{1}{N} \sum_{i=1}^{N} \varepsilon_{a_\alpha}^{i} \\[2ex]
SD = \sqrt{\frac{\sum_{i=1}^{N} (\varepsilon_{a_\alpha}^{i} - ME)^2}{N}}
\end{cases}
\tag{16}
$$

where $N$ is the number of grid cells in the domain and $\alpha$ can correspond to $TP$ ( total tracer precipitation or rain, snow or
graupel separately) or $TTPW$ (tracer total precipitable water).

An alternative statistical treatment, which is very visual and can be used as a second test of the reliability of the method, is that of Sodemann et al. (2009), based on computing the relative contribution of each moisture source to total precipitable water, total precipitation or to each type of precipitation (rain, snow or graupel) separately. The calculation returns the relative error of the mean values of those variables at each instant in time. For example, let $\overline{P}$ the mean total precipitation, then the
contribution (in %) of each source $S_n$ is:

$$
F_{TP_{S_n}} = 100 \cdot \frac{\overline{TP}_{S_n}}{\overline{P}}
\tag{17}
$$

where $\overline{TP}_{S_n}$ represents the mean total precipitation from source $S_n$. Then, if the method were perfectly accurate, the sum of all contributions ($\sum_n F_{TP_{S_n}}$) should equal 100%. The degree of deviation of this sum with respect to the latter value, yields the relative error ($\varepsilon_r$) of the mean tracer precipitation:

$$\varepsilon_{r_{\overline{TP}}} = \sum_n 100 \cdot \frac{\overline{TP}_{S_n}}{\overline{P}} - 100 = 100 \cdot \frac{\sum_n \overline{TP}_{S_n} - \overline{P}}{\overline{P}} \tag{18}$$

The above equation can be applied not only to total precipitation, but also to any particular type of precipitation or to total precipitable water. Finally, we note that the concept of relative error of the mean variables should not be confused with the mean relative error, which would be expressed, following the notation used in the equation above, as:

$$\overline{\varepsilon_{r_{TP}}} = \frac{1}{N} \sum_{i=1}^{N} \left( 100 \cdot \frac{\sum_n TP_{S_n} - P}{P} \right)_i \tag{19}$$

This last variable will also be used during the validation treatment shown below.

## 3.3 Validation results

As mentioned earlier, the validation experiment is a monthly long simulation for November 2014 over North America. Figure 4 shows the results obtained in this simulation for total precipitation from each of the five analyzed sources ($TP_{S1}$, $TP_{S2}$, …) depicted in Fig. 3, and the total sum of precipitation from all sources ($\sum_n TP_{S_n}$). The relaxation zone along the boundaries is excluded in these figures. The largest contribution to total precipitation is from external advection into the domain, and in the eastern half of it, also from sea evaporation. Lake evaporation is locally important around the Great Lakes and in Canada, where most smaller lakes in the grid are. Evapotranspiration over land is not very relevant in this month of November, and neither is its contribution to precipitation. Moisture present at initial time precipitates significantly only toward the eastern boundary of the domain, in the downwind direction of the dominant westerly flow.

According to Eq. (15), for the absolute error to be zero at each point, the result in Fig. 4f should exactly match the total precipitation calculated by the model, shown in Fig. 5a. The values of this error (i.e the differences between the results of Fig. 4f and Fig. 5a) are depicted in Fig. 5b. The maximum deviations between the sum of the precipitation coming from the five considered sources and the total precipitation calculated by the model, occur over the sea, near the domain's edges, and hover around -3 mm. These values correspond to very low relative errors (Fig. 5c), since the cumulative precipitation in these areas during the month of November is very high, often exceeding 300 mm. In most regions, however, the absolute error is clearly less than 1 mm, close to zero for the most part, thus, very small, even in the relative sense. Neglecting cells where the total monthly precipitation is less than 1mm to avoid arithmetical problems, the area-averaged value of the relative error (Eq. (19)) is -0.17%, with a standard deviation of 0.20%. The maximum relative error found at any point is only -3.73%, in areas of the US desert southwest with low accumulated precipitation during this month of November.

Figure 6 shows at 3h intervals, the mean error (ME) and standard deviation (SD) for the three precipitation types, rain (Fig. 6a), snow (Fig. 6c) and graupel (Fig. 6e), throughout the monthly period of simulation. Values of the mean error are

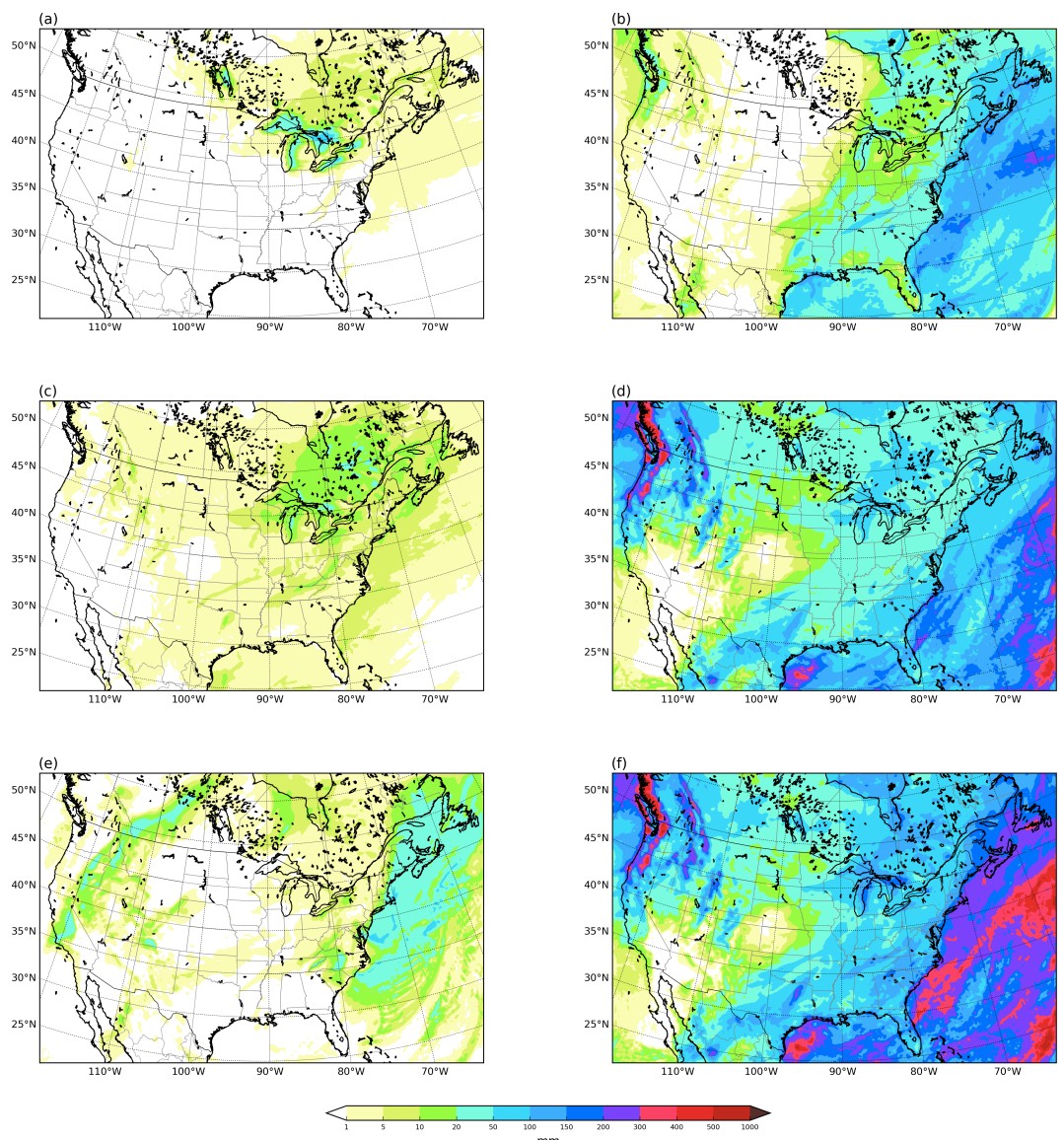

**Figure 4.** Total monthly accumulated tracer precipitation ($mm$): from lake evaporation (a), sea evaporation (b), land evapotranspiration (c), lateral boundary advection (d), initial moisture (e) and sum of all contributions (f).

very close to zero at all times, with small standard deviations of about 0.05 mm/day for rain, 0.01 mm/day for snow and 0.005 mm/day for graupel, indicating that the compensations between positive and negative errors are not very relevant. As expected, the error is larger for the domain-wide most abundant precipitation types (rain and snow, in this order) and smaller for the most residual type of precipitation (graupel). Bosilovich and Schubert (2002) found mean errors very close to zero for precipitation, as in our case, but comparatively much larger standard deviations of about 0.2 mm/day ($\sim 5\%$). In addition,

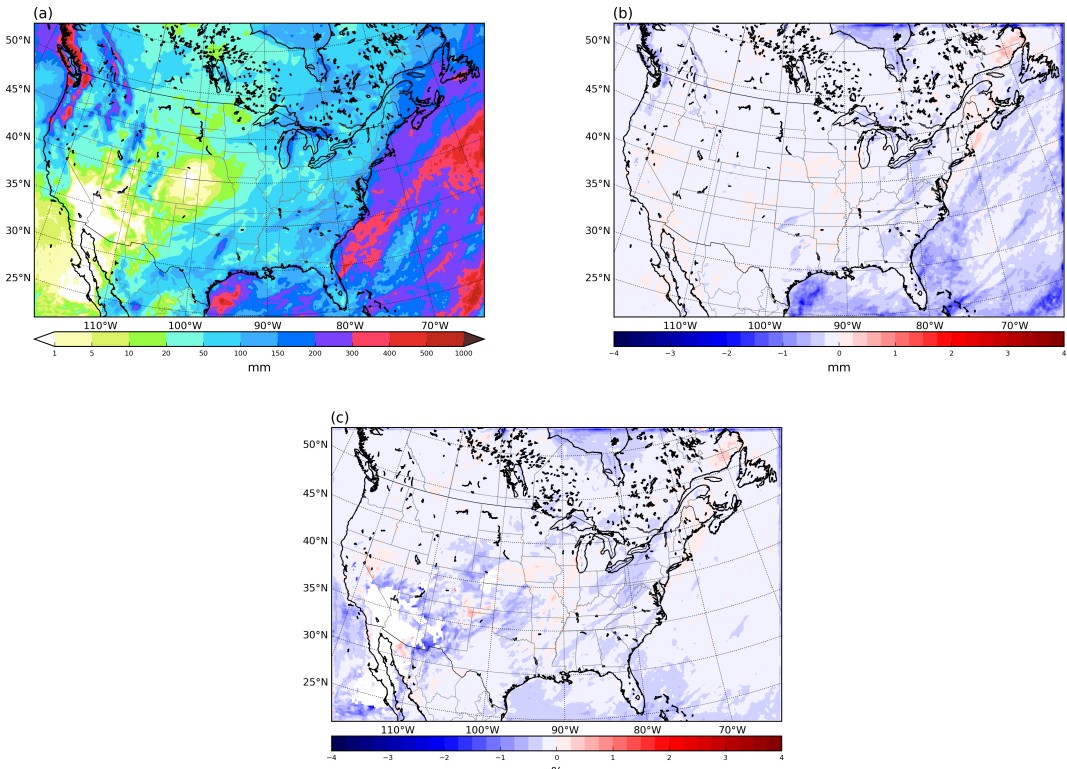

**Figure 5.** Total monthly accumulated model precipitation ($mm$) (a), tracer precipitation absolute error ($mm$) (b) and tracer precipitation relative error ($\%$) in areas where precipitation exceeds 1 mm (c).

Fig. 6 shows the relative contribution of each considered moisture source to area averaged rain (Fig. 6b), snow (Fig. 6d) and graupel (Fig. 6f). Moisture initially present in the domain's atmospheric columns only contributes to any precipitation type during approximately the first week of simulation. Rain is roughly about 40% of sea evaporation origin, and 60% from moisture influxes from the lateral boundaries, with this values oscillating throughout the month. In comparison with rain, snow and graupel have a stronger contribution from external moisture advection, and also from land evapotranspiration and lake evaporation, and much less fraction of sea evaporation input. As these figures are cumulative diagrams, the upper line (which separates the white zone from the color zone), indicates the combined contribution of all sources to precipitation. The deviation of this line from 100% represents the relative error of mean domain precipitation (Eq. (18)), which, as it is apparent, is very small for all three precipitation types and at all times. Further discussion will follow later in this section.

Validation results in terms of total precipitable water are presented in the diagrams of Fig. 7, which are similar to those in Fig. 6 for precipitation. In this case, the mean error (Fig. 7a) takes values around -0.01 mm, whereas the standard deviation is about 0.1 mm, which are very small numbers. To contextualize these results, we refer again to Bosilovich and Schubert (2002), who show a mean error around -0.5 mm ($\sim 2\%$) and standard deviation of about 0.5 mm.

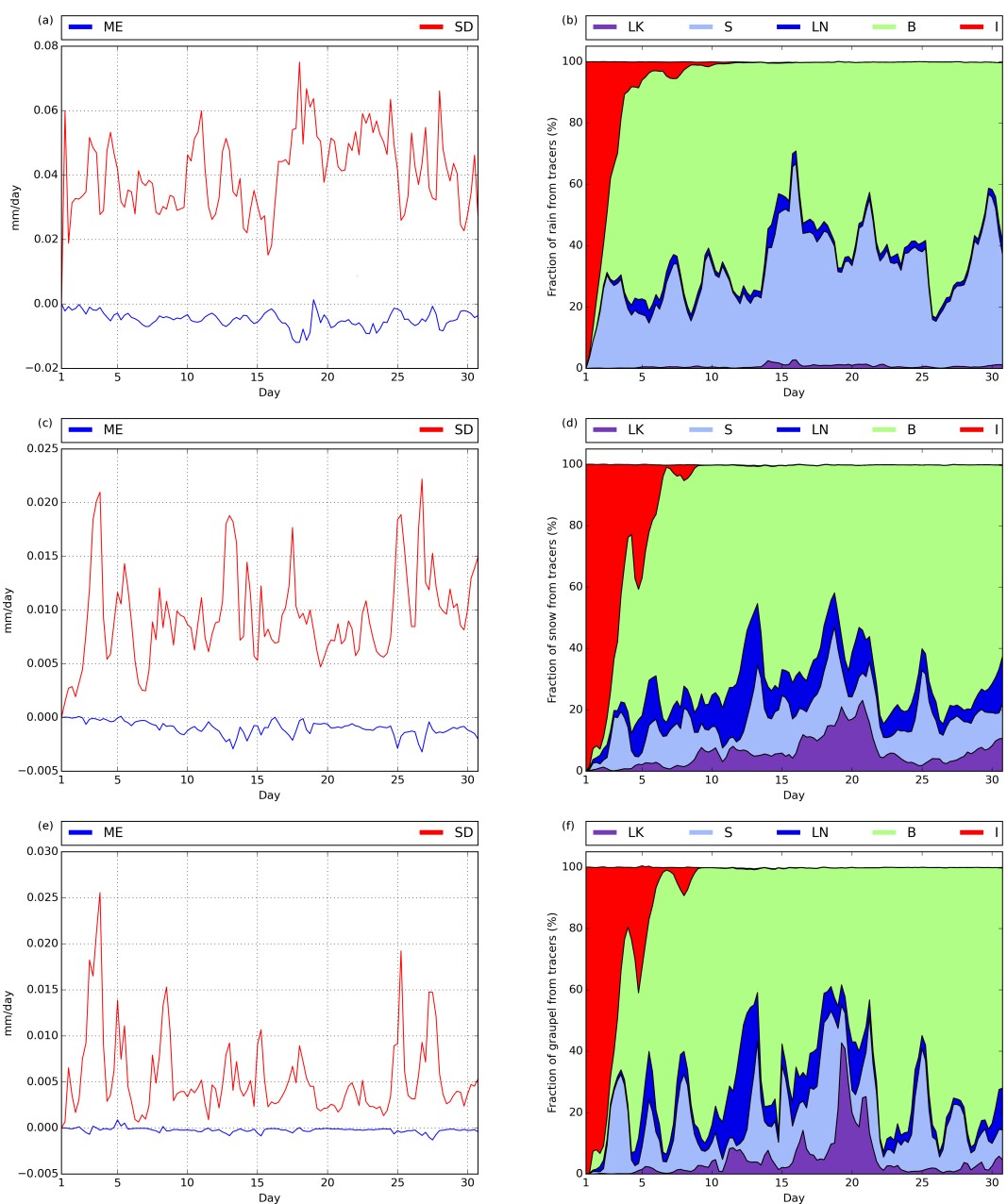

**Figure 6.** Mean error (blue) and standard desviation (red) $(mm)$ for 3h accumulated tracer rain (a), tracer snow (c) and tracer graupel (e). Relative contribution of each moisture source [lake evaporation (LK, purple), sea evaporation (S, light blue), land evapotranspiration (LN, dark blue), lateral boundary advection (B, green), initial moisture (I, red)] to 3h accumulated rain (b), snow (d) and graupel (f).

Finally, Fig. 8 shows in more detail the time evolution of the relative error of mean domain precipitation of all three types, as well as of mean domain TPW. This corresponds to the deviation from 100% in the cumulative values in figures Fig. 6b, d,

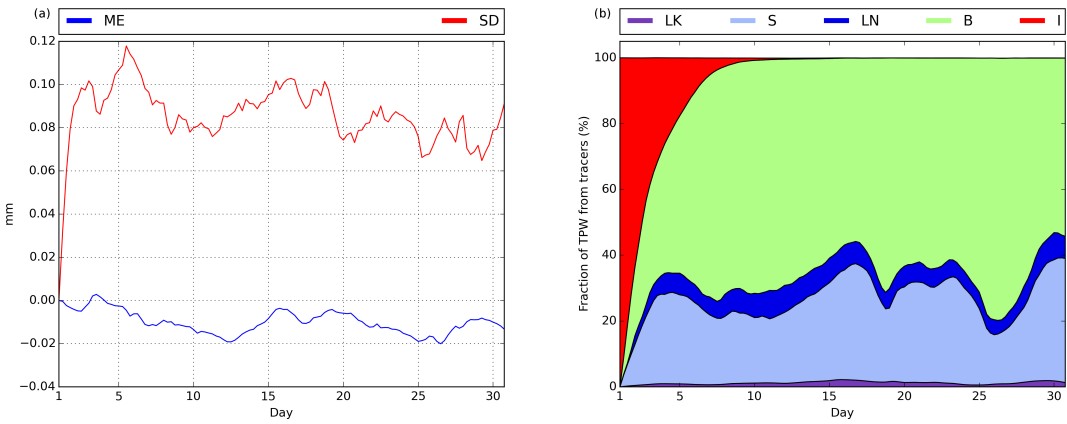

**Figure 7.** Same as Fig. 6 but for total precipitable water (TPW) ($mm$).

f and Fig. 7b, as discussed previously. Numbers are similar for the three precipitation types and do not exceed +/- $0.4\%$. On average, about $0.2\%$ of precipitation is not associated with any of the five considered moisture sources, i.e., the mean domain relative error is around $-0.2\%$. For TPW, errors are even smaller. In this case, the deviation of the sum of contributions from all sources from 100%, is roughly -0.1% (Fig. 8), which means that only 0.1% of TPW is not traceable. Sodemann et al. (2009) found, at first, errors that were around 10% for TPW, and later this value was improved to 1-2% (Sodemann and Stohl, 2013). Finally, we note that during the simulation period (one month) there is no increasing trend in these errors, which attests the method's stability.

Both the small absolute and relative values of the analyzed error measures in this section, together with the lack of trends in the errors, demonstrate the high accuracy and soundness of the method. Finally, with regard to the causes of these inaccuracies, most likely, they are largely caused by numerical errors derived from the very large moisture tracer gradients that occur in some regions of the domain, as for example, in the separation region between the "BOUNDARY" source (Fig. 3b) and the interior of the domain. These sharp transitions can induce small errors in the advection scheme and also stronger numerical diffusion than for full moisture. In addition, other errors, such as rounding errors or small inaccuracies in the water budget, contribute secondarily.

## 4  Application example: lake evaporation as moisture source in the Great Lakes snowstorm of 2014

Heavy snowstorms are common meteorological phenomena in the North American Great Lakes region during autumn and winter months, usually associated with the intrusion of a cold and dry polar air mass over the warmer lake waters (e.g. Wiggin, 1950; Eichenlaub, 1970; Hjelmfelt and Roscoe, 1983; Niziol et al., 1995; Wright et al., 2013). The resulting large water-atmosphere temperature contrast increases heat and moisture fluxes from the lakes, destabilizing the planetary boundary layer (e.g. Lenschow, 1973; Chang and Braham, 1991) and leading to an activation and/or intensification of precipitation downwind.

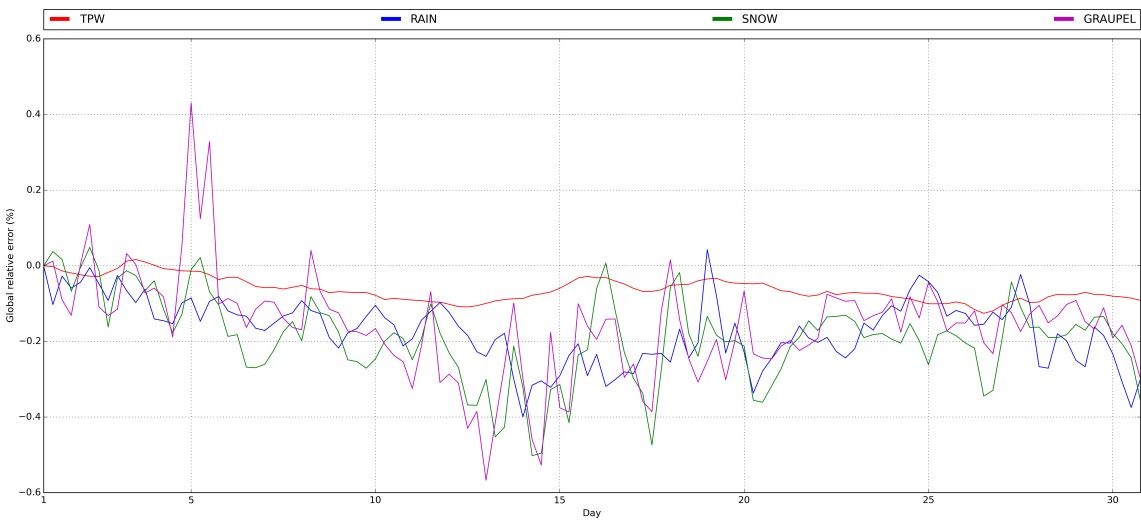

**Figure 8.** Relative error for mean domain tracer total precipitable water (TPW, red), 3h accumulated tracer rain (blue), tracer snow (green), and tracer graupel (purple).

In some occasions snow bands formed during these events produce huge snow accumulations, with high socio-economic impacts (e.g. Changnon, 1979; Eichenlaub, 1978; Schmidlin, 1993).

It is well established that heat and moisture fluxes from the lakes are fundamental in the development of these episodes, since they cease to occur once open waters freeze over. Given the low moisture content of polar air masses, it is also likely that without evaporative fluxes from the lakes, large accumulations of snow would not be possible. It is still not clear, however, what the actual input of lake water to snowfall is in these events. Studies about the contribution of evaporated moisture from the Great Lakes to precipitation in lake-effect snowstorms are scarce, based on the analysis of the isotopic composition of precipitation (the so-called physical moisture tracers) and do not correspond to particular extreme events but to climatic periods (Gat et al., 1994; Machavaram and Krishnamurthy, 1995). The WRF-WVT tool that we present here, can contribute to clarify this question, and, as an application example, in this section we quantify the role of the Great Lakes as moisture sources in the famous case of the November 2014 severe lake-effect snowstorm, the so called "Snowvember" by local residents, which affected especially New York state (mainly cities bordering lakes Erie and Ontario, and in particular, the Buffalo area) between the 17th and 21st of this month, causing at least 13 fatalities, widespread food and gas shortages due to blocked roads and, in general, many other traffic problems and material losses derived from the storm (National Weather Service, NOAA, 2014).

## 4.1 Experimental design

The example application experiment is run for four and a half days (17-00Z to 21-12Z November), in a D2 domain nested within the validation simulation and encompassing the Great Lakes region with a horizontal resolution of 5km and the same 35 vertical levels as the parent domain D1 (Fig. 2). Tracer moisture from the parent domain can feed the nested domain through

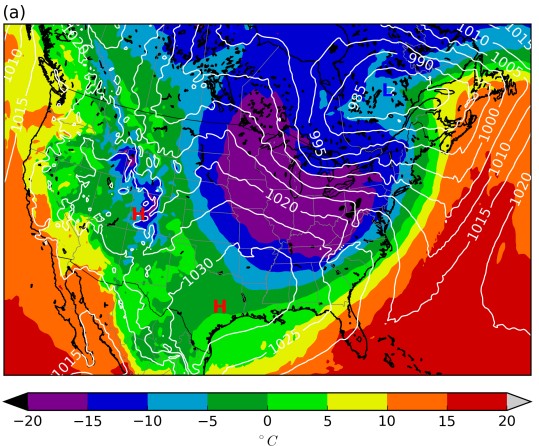 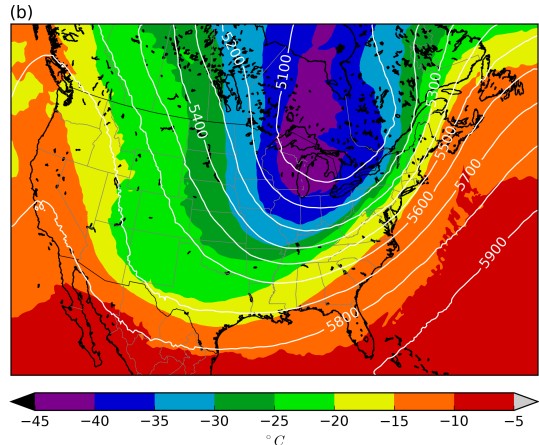

**Figure 9.** Synoptic situation on the 18th of November 2014 at 12UTC. Mean sea level pressure (contours, $hPa$) and 850hPa temperature (shades, $^oC$) (a). Geopotential height (contours, $m$) and 500 hPa temperature (shades, $^oC$) (b).

its lateral boundaries, which are not set to zero. The simulation serves also as an example of the versatility of the tagging tool. The physics settings in this experiment are identical to those in the validation simulation, except for spectral nudging and the convective parameterization, which are turned off.

Figure 9 shows the general synoptic situation for the selected case, in terms of surface pressure and 850 hPa temperature (Fig. 9a) along with 500 hPa geopotential height and temperature (Fig. 9b), both at 12 UTC on November 18th, 2014. The situation is the typically associated with Great Lake-effect snowstorms: a deep trough with low temperatures aloft over the region, causing intense west-northwest winds at lower levels across the Great Lakes and very cold air advection. The lakes were mostly ice-free at this time, with temperatures between 0-8$^oC$, the warmest in lake Erie (Fig. 10b), contrasting markedly with the below -15$^oC$ values at 850 hPa. The topography of the area is also show in Fig. 10, with the highest terrain east of lake Erie.

## 4.2 Results

## 4.3 Precipitable water

Figure 11 shows the daily evolution, from 17 to 20 November 2014, of the precipitable water originating from evaporation in the lakes and the 10m wind at 12 UTC. Paired panels depict the percentage of total precipitable water that those amounts represent, together with 850hPa temperature. At 12 UTC on the 17th, a short wave trough was pushing past the region. Winds ahead of the associated front were still from the south over lakes Erie and Ontario, with moderately low temperatures above -6$^oC$ at 850hPa; however, behind the trough, a very cold air mass was already in place over lakes Superior, Michigan and Huron, where winds had already veered and were at this time from the west-northwest direction. The enhancement of evaporation from the lakes is already apparent at this time, with precipitable water plumes from lakes Superior and Huron with values around

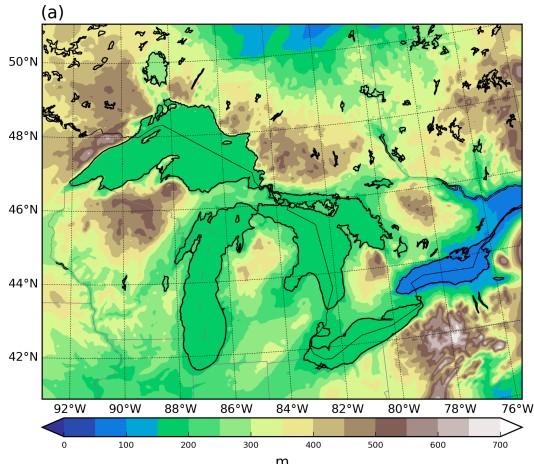
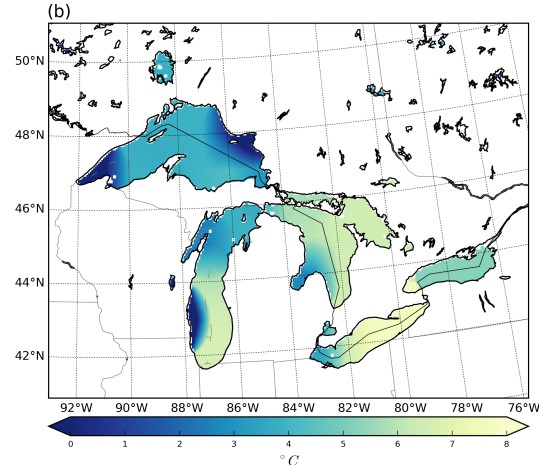

**Figure 10.** Topography of the nested domain ($m$) (a) and lake surface temperature of the Great Lakes ($^{o}C$) (b) on the 18th of November 2014 at 12UTC.

2-3 mm, which represent a contribution of $20-30\%$ of the total. After frontal passage, the next day winds increase in intensity and change direction to the west-northwest, and the cold air settles in with temperatures around -16$^{o}C$ at 850 hPa. The arrival of the cold and dry air mass, together with the wind intensity rise, augment evaporation fluxes from the surface of the lakes, so that the precipitable water with this origin practically doubles with respect to the previous day, increasing the lake moisture

contribution to about 30-60% of the total. The highest values are attained in plumes aligned with the main wind direction that originate from open waters and extend leeward of the lakes. The cold air stays in place for the next days and lake water evaporation values remain high; however, the direction of the moisture plumes from this source vary as wind changes due to the approach of another short wave trough, turning more toward the north as the flow becomes southerly on the 19th, and again westward of the lakes when winds turn in this direction on the 20th. In the areas where the 850hPa temperatures remain below

about -15$^{o}C$ during the short wave passage, plumes of moisture from the lakes still develop, with an input of lake moisture above 30% of total content.

### 4.4 Precipitation

The previous results suggest that the lakes' contribution to atmospheric moisture in the region is very significant for this event, and we assess next whether this is also the case for precipitation. Observed snowfall totals for the period between the 17th at

6UTC and the 21st at 6UTC (Fig. 12a, from NOAA's National Snow Analyses data, Carroll et al. (2006)) were very high, with peak values close to 100 mm in the Buffalo, NY area, to the lee of Lake Erie, and with other pockets of over 60 mm of snow water equivalent accumulations on the leeward shores of lakes Huron and Ontario, where orographic lifting from the existing hills further enhances precipitation. Figure 12b shows model results for the same period, which are in very good agreement with the observations, in amounts and distribution. This is particularly true for the aforementioned areas of highest snowfall

totals.

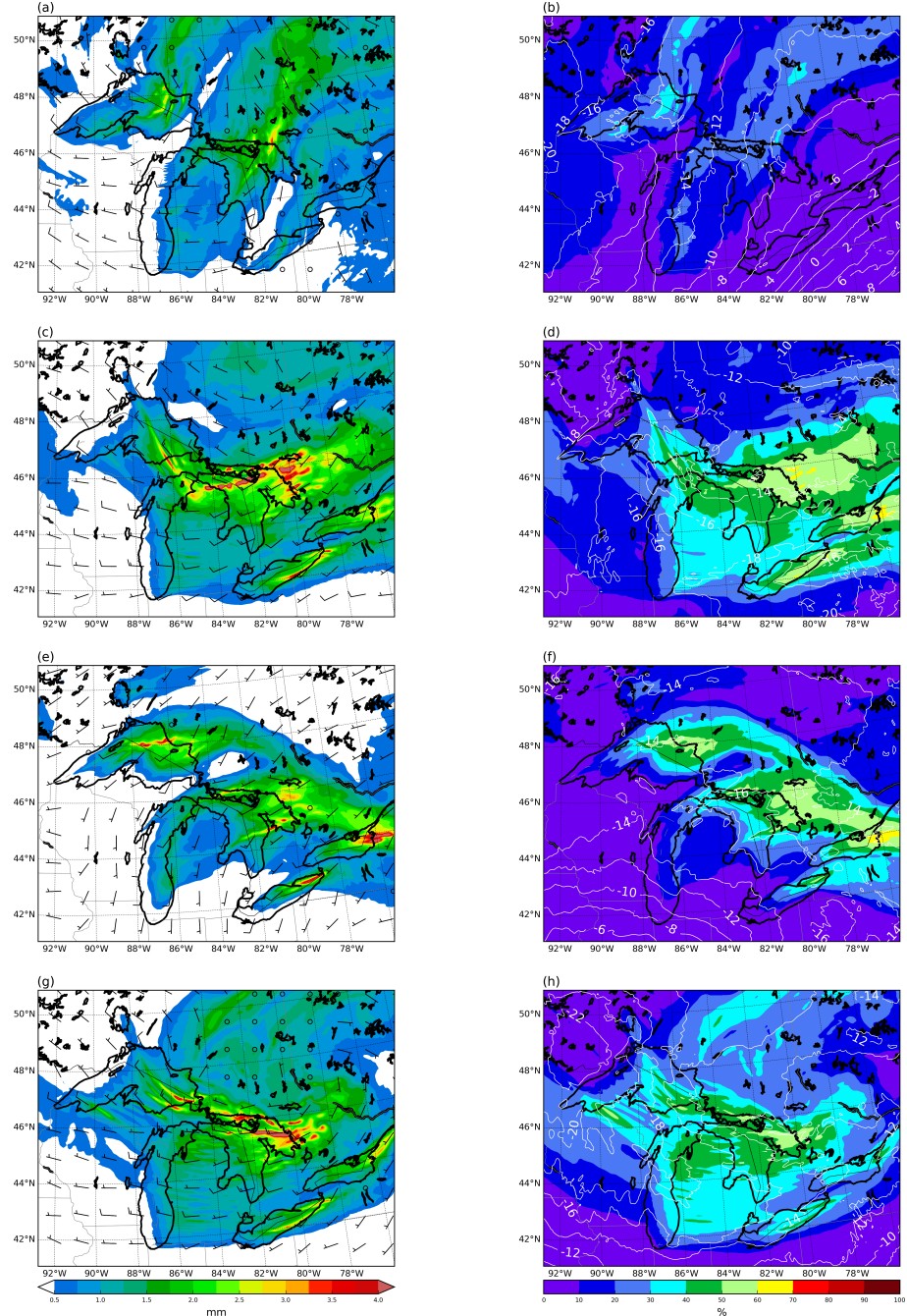

**Figure 11.** Total precipitable water ($mm$) originating from lake evaporation on the 17th (a), 18th (c), 19th (e), 20th (g) of November 2014 at 12Z and their percentage contribution to total precipitable water for the same times (b, d, f, h). Barbs show 10m winds and contours 850hPa temperature ($^oC$).

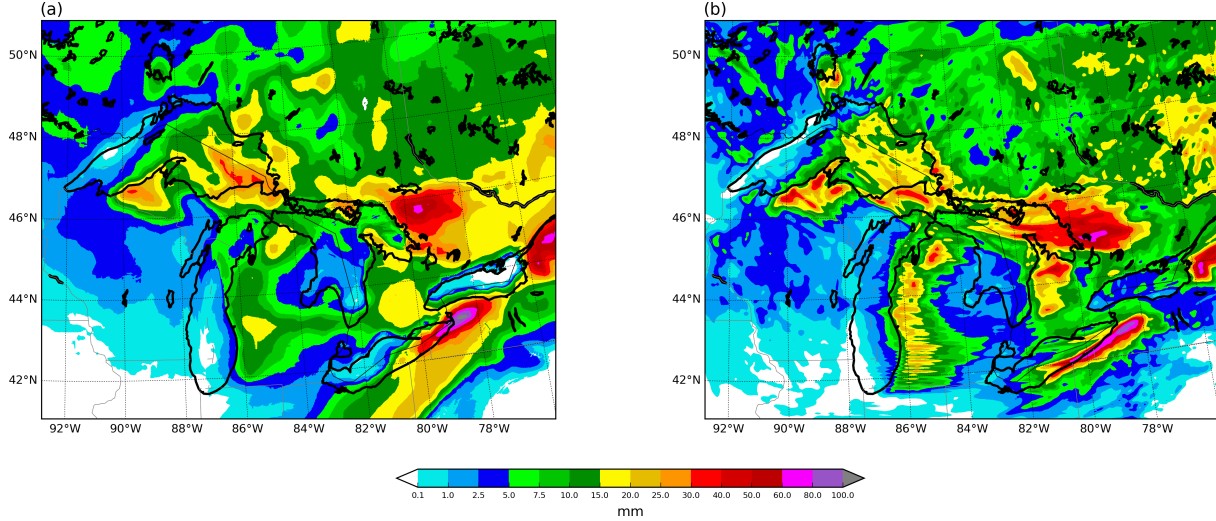

**Figure 12.** Observed (a) and simulated (b) accumulated snow water equivalent ($mm$) from November 17th at 6UTC to November 21st at 6UTC.

The part of precipitation originating from lake evaporation during the same four-day period is shown in Fig. 13, in terms of absolute (Fig. 13a) and relative (Fig. 13b) values to total accumulations. The role of the lakes as moisture sources is very relevant. In general, in all regions immediately downwind of the Great Lakes, water vapor with this origin accounts for more than 30% of precipitation. The areas where the contribution of lake water vapor fluxes to precipitation is largest coincide with the locations of maximum snowfall totals, to the lee of lakes Huron, Erie and Ontario. Here, more than 50% percent of the snow water equivalent has its source in lake evaporation, which attests the fundamental role of lake moisture in producing the observed localized extreme accumulations during these events. In regions further from the lakes, the pattern of total precipitation and that of precipitation originating from lake evaporation lose correlation.

## 5    Summary and conclusions

We presented here a new moisture-tagging tool, coupled to the WRF model V3.8.1, for the analysis of precipitation sources and atmospheric humidity pathways in general. The technique is framed within the online Eulerian methods usually known as water vapor tracers (WVTs). We first detailed the method's formulation and its implementation into WRF, which required the modification of the turbulent, the microphysics and the cumulus parameterizations for the calculation of the associated tracer tendencies. We then assessed the method's precision with a validation strategy consisting in tagging moisture from all possible sources and evaluating the difference between the sum of all these contributions and total moisture results, in terms of precipitable water and precipitation. We identified the method's error with these deviations. The sources considered were: incoming fluxes from the model grid's lateral boundaries, the moisture initially present in the entire atmospheric volume of the domain, and surface evaporation. We further divided evaporative sources into three, namely ocean, land and lakes, which

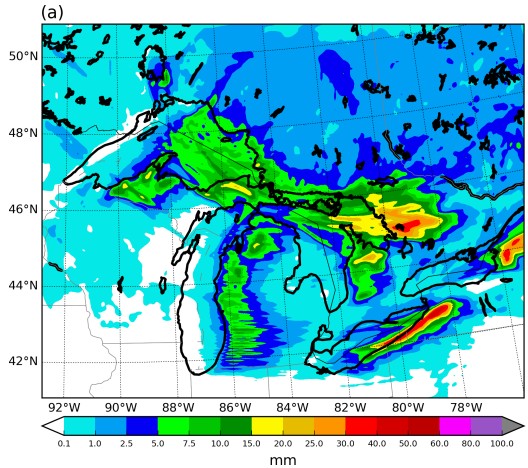 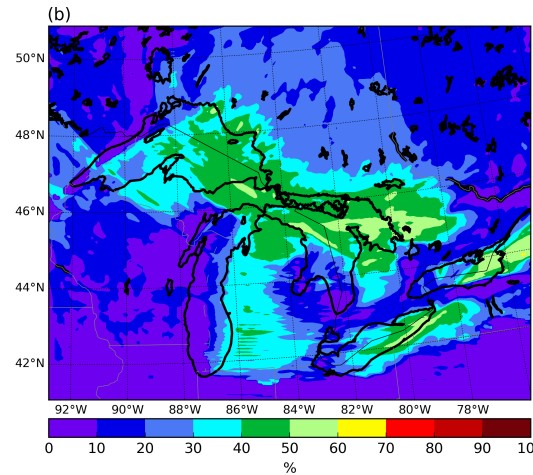

**Figure 13.** Simulated accumulated tracer snow water equivalent (i.e., coming from the lakes evaporation) ($mm$) (a) and its percentage contribution to total simulated accumulated snow water equivalent (b) from November 17th at 6UTC to November 21st at 6UTC.

made the validation somewhat more challenging. We performed a one-month long (November 2014), continental scale (North America), 20km resolution model simulation for this purpose, and found that the deviations of area-averaged-variables are consistently about -0.1% for precipitable water and -0.2% for 3h accumulated rain, snow or graupel. This means that there is a small amount of precipitable water and precipitation that the method cannot link to any source. There is no noticeable increasing

trend in these errors during the monthly long period of simulation. The mean relative error and the standard deviation for the monthly-accumulated precipitation is -0.17% and 0.2%, respectively, about the same as for the 3h values throughout the same period. These results demonstrate the robustness of our WRF-WVT implementation as a sound and highly accurate tool to track atmospheric moisture pathways.

     Finally, as an example application of the moisture tagging technique, we simulated the Great-Lake effect snowstorm of 2014,

aiming at quantifying the contribution of evaporative fluxes from the lakes to total precipitable water, and especially to snowfall amounts in this event. We employed for this purpose a nested grid within the validation domain, covering the Great Lakes region at 5km resolution and simulated the four-day period from November 17 at 6UTC to 21 at 6UTC. Results show the activation of the lake effect upon arrival of a cold and dry arctic air mass over the area, with the formation of total precipitable water plumes originating from the lakes and extending tens and even hundreds of kilometers in the downwind direction. As expected,

the model shows how the lake effect intensifies with colder and stronger west or northwesterly surface winds and tampers out with warmer and weaker southerly airflows. The contribution of lake evaporated moisture to total precipitable water within the plumes is generally above 30% across the area downwind of the lakes when temperatures at 850 hPa are below around -15$^{o}C$, and exceeds 60% in plumes to the lee of lakes Huron, Ontario and Erie when conditions are most favorable for lake-effect, on the 18th of November.

The model simulation reproduces faithfully observed snowfall accumulations during the four-day period, with maximum amounts of close to 100 mm of snow water equivalent in the Buffalo, NY area, to the lee of lake Erie, and other pockets with

values above 60 mm on the leeward shores of lakes Huron and Ontario. It is in these locations of highest impact where the contribution of lake evaporation to precipitation is largest, between 50-60% of the total. In general for all regions immediately downwind of the lakes, the input of lake moisture to precipitation is about 30-50%, and diminishes gradually at further distances.

These results highlight the important contribution of evaporative moisture fluxes from the lake surfaces in the genesis of precipitation during Great Lake-effect snowstorms. They also suggest that this input is fundamental in producing the most extreme accumulations, with the highest socio-economic impacts, in the Buffalo, NY area and other locations to the lee of the lakes, especially Erie, Ontario and Huron. To draw a more robust general conclusion, an in-depth investigation with a sufficient number of cases and further diagnostics would be needed; however, this is beyond the scope of the present article and a matter

of future work, since our intent here is to simply illustrate with a practical example the possibilities of WRF-WVT as a powerful tool for moisture tracking.

*Competing interests.*  Authors declare that no competing interests are present.

*Acknowledgements.*  Funding comes from the European Commission FP7 (EartH2Observe) and from the Spanish Ministerio de Economía y Competitividad (CGL2017-89859-R and CGL2013-45932-R). Computation is at CESGA (Centro de Supercomputación de Galicia), Santi-
ago de Compostela, Galicia, Spain.

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
