# Peer review of "A new moisture tagging capability in the Weather Research and Forecasting Model: formulation, validation and application to the 2014 Great Lake-effect snowstorm"

_Earth System Dynamics, 2017_

## Referee Comment (RC1) · Anonymous Referee #1 · 26 Oct 2017

Dear Editor,

I have read and assessed the manuscript, which describes the implementation of online moisture tracking in the WRF atmospheric model and an application of this model to winter snowfall over the US Great Lakes area. The subject of the manuscript is very relevant as indicated by the many alternative methods to track atmospheric moisture. Moreover, the manuscript is well written.

[Figure]

The scientific quality of the work is good. However, my main issue with is that a validation has been performed in terms of the internal moisture budget of the method. I am happy that the errors in this budget are close to zero, but would really be interested how this new model performs compared to alternative methods, although I realize this would be a substantial effort.

Specific comments:

- In the Introduction, the author give an overview of different moisture tracking models (Eulerian, Lagrangian, on-line and off-line) and the assumptions that are associated with some of the current implementations of these models. In P2L25 it is stated that "Lagrangian models include, ..." as if this is true for the entire class of Lagrangian models, whereas I think it is only true of the implementations mentioned. That is, it is perfectly possible to create an off-line Lagrangian model that does not have these drawbacks. Therefore, I would encourage the authors to rewrite this section and to state clearly whether the assumptions are a limitation of the method or of the implementation.

- P2, L33-P3,L2: Here the authors state that the sub-grid variability in vertical motion is a drawback of Lagrangian models. Is this not true for all off-line simulation, so also for Eulerian models?

- P2L27, "simplifications that each author assumes". Again not clear whether these are method specific or implementation specific. Maybe this is a good place for a table of moisture tracking methods?

- P7L8-9: "but not all are equally treated". Can you state in which way the forms are simplified? And how does this relate to the validation later on?

- P8L12: This is really an internal validation of the system, in the sense that the budgets should match. Therefore, maybe "Moisture tracers budget validation" is a title?

- I had really hoped for a comparison between the offline moisture tracking schemes

and the model in this paper. How does the new technique relate to the moisture recycling estimates from offline schemes? Without this comparison, the reader does not really know whether to switch to an online tracking scheme, or use the offline scheme, which is much easier to run. See for example van der Ent et al(2013) for such a comparison.

Reference: Van der Ent, R. J., Tuinenburg, O. A., Knoche, H. R., Savenije, H. H. G., & Kunstmann, H. (2013). Should we use a simple or complex model for moisture recycling and atmospheric moisture tracking?. Hydrology and Earth System Sciences Discussions, 10 (5), 2013.

- P15L1-6: So, where do these errors come from? Numerical stability issues? Precision (rounding) issues?

- Related to that the budget errors: how do these errors compare to the moisture budget of the model? Is that zero, or is moisture missing there as well?

- The shading in Figure 10b looks very strange, with very large temperature gradients. What data is used for this, how is it interpolated and on what resolution?

- P19L32-33: "the pattern ... correlation": unclear, please rephrase.

- P21L8-10: "This means ... source": unclear, please rephrase.

- P21L31: "important contribution of evaporative fluxes": How can you be certain that it is the evaporation? Is this effect isolated from any temperature effects? If so, how is it determined?

- P21L35: "further diagnostics": Can you state what kind of diagnostics? Any ideas?

Minor comments: - P1L6: "monthly" –> "a one month"

- P7L8: could –> cloud

- P19L1: "first ... region": Unclear what is meant with this sentence, maybe rephrase it

so it is clearer.

- P19L8-9: "flow of moisture from the surface" –> evaporation?

- P21L24: "18th" –> "18th of November"
* * *

---

## Referee Comment (RC2) · Anonymous Referee #2 · 30 Oct 2017

After reading the manuscript and the interactive comments by the other anonymous referee, I can say that my opinion about the paper submitted by Damian Insua-Costa and Gonzalo Miguez-Nacho is highly positive.

[Figure]

The origin of moisture to produce precipitation in a particular region is a very important meteorological problem. The authors of the present paper review some available methods of moisture origin assignation to the observed or modelled precipitation and/or precipitable water in their paper, and they propose a new method that they have incorporated to the WRF model. They validate the method through a month of integration (over US) and they apply it to analyse the interesting US Great Lakes snowstorm of November 2014. The paper is not only a good contribution to the main meteorological problem already mentioned, as well to the understanding of a very interesting particular case, but it is also a very well written paper, clear and with well-presented complementary figures.

I would accept the paper for publication almost as it is, although the comments of referee #1 can surely improve the text. I would only add a few small complementary details: Noting that the method is intrinsically coherent from the modelling point of view (the error of the addition of all the contributing origins into the total modelled precipitation is very small), it is worthy to compare the observed and modelled precipitation, in order to better evaluate the significance of the possible contribution of the different moisture origins to the observed/actual precipitation:

Fig. 12 does compare the observed and modelled total precipitation for the case of November 2014 in Great Lakes; why do not do it (in Fig. 5) to compare the observed and modelled precipitation (only on land, of course) during the whole validation month?

Although it is clear in the text, perhaps in Fig. 13 it would be convenient to specify that the amount and percentage of precipitation represented in it is the part which origin is the Great Lakes evaporation.

Pg. 11, line 11: a mistake, 2104 (2014)

---

## Referee Comment (RC3) · Anonymous Referee #3 · 30 Oct 2017

In this paper, the authors develop an online moisture tracking scheme within the WRF model, and validate the performance of the model against observations of an extreme winter precipitation event in the Great Lakes region of the United States. I recommend publication pending minor revisions (see below).

GENERAL COMMENTS

In general this is a very good paper, with an excellent background on the range of mois-

ture tracking options available including Lagrangian and Eulerian tracking schemes, as well as a detailed explanation of the model, data sources, and the novel contributions of this team. The moisture tagging approach that is implemented within WRF is somewhat outside of my academic background, but in general appears sound. The validation of the approach against an observed extreme event is particularly interesting and it is a sign of how far the science has come in recent years, especially in realistically representing surface and atmosphere coupling during extremes. I have only a few minor comments:

1. First in the introduction, the authors briefly discuss offline Eulerian tracking schemes, and it is suggested that the vertical integration is a significant shortcoming of these approaches. I think that if the authors are going to cite the Goessling and Reick (2013) paper (which is critical of the single column version of the WAM-2layers, as described in Keys et al. 2012), the authors should also cite how this issue has been addressed using a two-layer, model-level version of the Eulerian tracking scheme, which performs favorably relative to regional climate model comparisons. One of the other reviewers already highlights this, by pointing the authors to van der Ent et al. (2013) "Should we use a simple or complex model for moisture recycling and atmospheric moisture tracking?" https://doi.org/10.5194/hess-17-4869-2013

Other work that has used the two-layer, model-level tracking scheme includes van der Ent et al. (2014) which couples the WAM-2layers to a land-surface hydrology model, Keys et al. (2014) which examines whether the WAM-2layers can be used with multiple datasets, and Duerinck et al. (2016) which examines soil moisture coupling in Illinois. I am by no means suggesting the authors cite this list of other papers, but rather am illustrating that much work has been done to address the single column assumption, and now in using the improved version.

I do recommend the authors consider adding a sentence or two more at Page 2, Line 17 to more accurately represent the current state of Eulerian tracking generally (and that as a 'class' of tracking schemes some Eulerian models have addressed the valid

criticism associated with the single model level integration).

2. The authors make a point in the final sentence of the abstract by writing "...resulting in the highest socio-economic impacts." Since this is the final sentence in their abstract I think the authors ought to either:

a) explore this a bit more, clarifying what those socio-economic impacts actually were (in specific terms) during the snowstorm event, which populations were affected, and maybe even the adequacy of alerts and warnings ahead of the snowstorm.

b) eliminate any reference to that aspect of the paper.

I think that the authors have done such an amazing job with the rest of this work that it seems a little bit like they are doing themselves and the reader a disservice by mentioning socio-economic impacts so blithely (aside from the mention at Page 15, 1st paragraph of section 4). I think it is the norm in this field to feel obligated to say something about socio-economic impacts since you have to justify why this science matters. At this point, if the justification is 'socio-economic impacts' then I'm not convinced that this science helps with anything. I think it could, such as through improved monitoring of lake temperatures, regional humidity, etc. and coupling such monitoring insights with emergency management and weather monitoring stations. Perhaps this was already done during the snowstorm. But I think that the authors ought to dig a bit deeper here, if they want to justify the paper as such.

MINOR CORRECTIONS (Page = P, Line = L)

P1 L21 Change 'especial' to 'special'.

P3 l2 Check formatting for the citation.

P5 L7 The last clause of this sentence is confusing; consider revising for clarity.

P11 L11 Change '2104' to '2014'.

P11 L21 Change 'precipitations' to 'precipitation'

P15 L11 Good overview of the snowstorm event, but this is not adequate for justifying this work. Consider adding more substantive context for using this storm as a justification for the approach (perhaps in the summary section, or wherever is appropriate).

P16 L5 Cite the source of the "Snowvember" reference.

P16 L17 Change 'Eire' to 'Erie'.

REFERENCES

Duerinck, H. M., van der Ent, R. J., van de Giesen, N. C., Schoups, G., Babovic, V., & Yeh, P. J. F. (2016). Observed Soil Moisture–Precipitation Feedback in Illinois: A Systematic Analysis over Different Scales. Journal of Hydrometeorology, 17(6), 1645-1660.

Gößling, H., & Reick, C. H. (2013). On the" well-mixed" assumption and numerical 2-D tracing of atmospheric moisture. Atmospheric Chemistry and Physics, 13, 5567-5585.

Keys, P. W., Barnes, E. A., van der Ent, R. J., & Gordon, L. J. (2014). Variability of moisture recycling using a precipitationshed framework. Hydrology and Earth System Sciences, 18(10), 3937.

Keys, P. W., Van der Ent, R. J., Gordon, L. J., Hoff, H., Nikoli, R., & Savenije, H. H. G. (2012). Analyzing precipitationsheds to understand the vulnerability of rainfall dependent regions. Biogeosciences, 9(2), 733-746.

Van der Ent, R. J., Tuinenburg, O. A., Knoche, H. R., Savenije, H. H. G., & Kunstmann, H. (2013). Should we use a simple or complex model for moisture recycling and atmospheric moisture tracking?. Hydrology and Earth System Sciences Discussions, 10 (5), 2013.

Van der Ent, R. J., Wang-Erlandsson, L., Keys, P. W., & Savenije, H. H. G. (2014). Contrasting roles of interception and transpiration in the hydrological cycle-part 2: Moisture recycling. Earth System Dynamics, 5(2), 471.

---

## Author Comment (AC1) · 19 Dec 2017

I have read and assessed the manuscript, which describes the implementation of online moisture tracking in the WRF atmospheric model and an application of this model to winter snowfall over the US Great Lakes area. The subject of the manuscript is very relevant as indicated by the many alternative methods to track atmospheric moisture. Moreover, the manuscript is well written.

Thank you very much for your review. We believe that the modifications you suggest will improve the manuscript. Please, find below the responses to your comments.

The scientific quality of the work is good. However, my main issue with is that a validation has been performed in terms of the internal moisture budget of the method. I am happy that the errors in this budget are close to zero, but would really be interested how this new model performs compared to alternative methods, although I realize this would be a substantial effort.

We are also interested in comparing the results provided by the WVTs tool with those provided by other methods. However, we consider that this matter should be dealt with in a separate article. A comparison with alternative methods cannot be considered a validation of the model because there is no ground truth as reference; there are no moisture tracers' measurements, and even if there were, such as estimates of the content of certain isotopes in precipitation, there would always be an inherent uncertainty due to errors in the WRF model results. The tracers can only be validated internally, that is, within the world of the meteorological model. In any case, if the internal validation, which is not the same as budget validation (see below) yields good results and we know that the WRF model works correctly, we can affirm with a high degree of certainty that the method is realistic. In summary, this paper aims to be exclusively a presentation and validation of this method; we would not want to distract the reader's attention by introducing results provided by alternative methods.

Finally, we would like to mention that we are currently working on this issue and it is likely that in a short time we will have a publication where a comparison with other methods is discussed in detail.

Specific comments:

- In the Introduction, the author give an overview of different moisture tracking models (Eulerian, Lagrangian, on-line and off-line) and the assumptions that are associated with some of the current implementations of these models. In P2L25 it is stated that "Lagrangian models include, ..." as if this is true for the entire class of Lagrangian models, whereas I think it is only true of the implementations mentioned. That is, it is perfectly possible to create an off-line Lagrangian model that does not have these drawbacks. Therefore, I would encourage the authors to rewrite this section and to state clearly whether the assumptions are a limitation of the method or of the implementation.

We agree with the reviewer that some statements we made may be misleading. For this reason, we are going to rewrite this section and clarify some details related to these drawbacks. However, although some of the hypotheses formulated above could be relaxed, we do not know any Lagrangian model that completely avoids these inaccuracies.

- P2, L33-P3,L2: Here the authors state that the sub-grid variability in vertical motion is a drawback of Lagrangian models. Is this not true for all off-line simulation, so also for Eulerian models?

Yes it is. We will clarify this in the text.

- P2L27, "simplifications that each author assumes". Again not clear whether these are method specific or implementation specific. Maybe this is a good place for a table of moisture tracking methods?

Following the advice of the reviewer, the sentence "simplifications that each author assumes" is going to be replaced by "simplifications that authors assume in their own implementations". In addition, a table with the different moisture tracking methods will be added here.

- P7L8-9: "but not all are equally treated". Can you state in which way the forms are simplified? And how does this relate to the validation later on?

Here, we wanted to summarize with a single equation the different variations in moisture species due to the convection phenomena. This is difficult to find in the literature and we follow the formalism in Bechtold et al. (2001) to give a general expression. However, the Kain and Fritsch parameterization we use, does not apply this equation to all species equally. Most notably, entrainment processes are assumed to occur only for water vapor, that is, only environmental water vapor and not any of the other hydrometeors is mixed within the forming updraft and downdraft in the parameterized convective cloud. Furthermore, not all six moisture species in the WSM6 microphysics scheme are considered individually, since there are no equations for graupel. We have not made any simplification, what we meant is that the Kain and Fritsch convection parameterization uses simplified forms of the general equation to treat some of the moisture species. It is merely a question of mathematical formalism and therefore, this has nothing to do with the validation results.

Bechtold, P., Bazile, E., Guichard, F., Mascart, P., and Richard, E.: A mass-flux convection scheme for regional and global models, Quarterly Journal of the Royal Meteorological Society, 127, 869–886, https://doi.org/10.1256/smsqj.57308, 2001.

- P8L12: This is really an internal validation of the system, in the sense that the budgets should match. Therefore, maybe "Moisture tracers budget validation" is a title?

We agree with the reviewer in that it is really an internal validation of the system. However, we do not find the term "Moisture tracers budget validation" completely right because the validation does not only take into account the water budget. For example, it could happen that the water budget was perfectly closed but the validation poor if the moisture redistribution was not correct. A simple way to test this would be for example to switch off turbulence, convection and microphysics for tracers, and just leave redistribution by advection. Tracer

amounts would be conserved and the tracer's moisture budget would certainly be closed, however yielding very inaccurate results in terms of redistribution. There would be no precipitation from tracers. In some locations, there would be more tracer than full moisture. It is not straightforward to obtain that moisture from different sources be conserved and redistributed correctly so that, when we add the contribution of all sources to the precipitation, it coincides with total precipitation. That the moisture budget is conserved is part of the problem, but the key is that redistribution is correct, or in other words that the method tracks moisture accurately. For this reason, we think that the strategy that we follow for validation really assesses the traceability aspect of the method and not just moisture conservation, so we prefer to keep the title of the section as it is.

- I had really hoped for a comparison between the offline moisture tracking schemes and the model in this paper. How does the new technique relate to the moisture recycling estimates from offline schemes? Without this comparison, the reader does not really know whether to switch to an online tracking scheme, or use the offline scheme, which is much easier to run. See for example van der Ent et al(2013) for such a comparison.

Reference: Van der Ent, R. J., Tuinenburg, O. A., Knoche, H. R., Savenije, H. H. G., & Kunstmann, H. (2013). Should we use a simple or complex model for moisture recycling and atmospheric moisture tracking?. Hydrology and Earth System Sciences Discussions, 10 (5), 2013.

We totally agree with the reviewer on the need to compare with offline models. However, as we discussed above, we believe that this should be done in a separate article. As already mentioned, we are working on it but we still do not have conclusive results. We hope we can answer these questions soon.

- P15L1-6: So, where do these errors come from? Numerical stability issues? Precision (rounding) issues?

Most likely, the deviations are probably caused by numerical errors derived from the very large moisture tracer gradients that occur in some regions of the domain, as for example, in the separation region between the "BOUNDARY" source (Fig. 3b) and the interior of the domain. These sharp transitions can induce small inaccuracies in the advection scheme and also stronger numerical diffusion than for full moisture. In addition, there are also rounding errors.

We will now add a sentence commenting these possible sources of error in the revised text.

- Related to that the budget errors: how do these errors compare to the moisture budget of the model? Is that zero, or is moisture missing there as well?

The figure below shows the water budget terms for the full simulation period, accumulated daily in mm, where E is evaporation, P precipitation, Flux is the net flux though the lateral boundaries, and dW/dt the variation in moisture store. RES corresponds to the residual, the error in the water budget. All terms are area averaged for the whole domain, excluding the relaxation zone along the lateral boundaries, where moisture is not conserved. Errors in the budget are very small, less than 0.1 mm/day on average. They are however larger than the mean errors for the tracer method, shown in Fig 6. Errors in the water budget are related to the formulation of the model in general, and therefore affect tracer calculations as well as full

moisture. While it is not guaranteed that these errors present a linear behaviour, it is likely that they cancel out somehow when comparing the full moisture budget with those of tagged moisture from different origins, thereby yielding the very small values obtained in our calculations of the tracer method error.

We will add a comment on the relative size of the errors presented for the tracer method and those of the general water budget of the model.

[Figure]

- The shading in Figure 10b looks very strange, with very large temperature gradients. What data is used for this, how is it interpolated and on what resolution?

This figure represents the SST (sea surface temperature) input variable, which is updated every 6 hours and linearly interpolated for the timesteps in between. Data is from ERA-Interim reanalysis (just as for the rest of the input variables), and it's on a global Gaussian grid of approximately 70km resolution. It is interpolated to the model grid with a linear method. The gradients may seem larger than they really are because the temperature scale is quite short (from 0ºC to 8ºC) and due to the colour choice. We will now change the shading in the figure, so that transitions are not so abrupt.

[Figure]

(b)

- P19L32-33: "the pattern ... correlation": unclear, please rephrase.

The sentence "In regions further from the lakes, the pattern between total precipitation and precipitation from the lakes loses correlation" is going to be replaced by "In regions further from the lakes, the pattern of total precipitation and that of precipitation originating from lake evaporation, lose correlation".

- P21L8-10: "This means ... source": unclear, please rephrase.

The sentence "This means that there is a small amount of precipitable water and precipitation of equal-to-the-latter absolute values that the method cannot link to any source" is going to be replaced by "This means that there is a small amount of precipitable water and precipitation that the method cannot link to any source".

- P21L31: "important contribution of evaporative fluxes": How can you be certain that it is the evaporation? Is this effect isolated from any temperature effects? If so, how is it determined?

As we demonstrate in the study, a significant fraction of precipitation originates from evaporation in the lakes. The tracer method allows for the isolation of the evaporative contribution, which we can accurately quantify in a realistic model simulation without interfering at all with the thermodynamics of the event. The whole paper is about how we determine the fraction of precipitation from a given source at any location using WRF. We believe that it is therefore correct and well supported to state in the conclusions that "These results highlight the important contribution of evaporative moisture fluxes from the lake surfaces in the genesis of precipitation during Great Lake-effect snowstorms".

- P21L35: "further diagnostics": Can you state what kind of diagnostics? Any ideas?

For example, if a sufficient number of cases were analyzed, we could study whether the contribution of lakes (evaporation) to this type of snowfall has increased, decreased or maintained in recent years due to changes in the lakes' temperatures or a general warming of air temperatures. We could also study the relation between the size of the evaporative contribution of the lakes and the air-water temperature contrast. With this sentence we are referring to the many possible different aspects that can be investigated to characterize the contribution of evaporation from the lakes to precipitation in Great lake effect snowstorms that go beyond merely quantifying the precipitation fraction with this origin.

-P1L6: "monthly" –> "a one month"

The sentence will be corrected.

- P7L8: could –> cloud

The typo will be corrected.

- P19L1: "first ... region": Unclear what is meant with this sentence, maybe rephrase it so it is clearer.

The sentence will be rephrased as: "a short wave trough was pushing past the region".

- P19L8-9: "flow of moisture from the surface" –> evaporation?

The sentence will be changed to: "augment evaporation fluxes from the surface of the lakes".
- P21L24: "18th" –> "18th of November"

The sentence will be corrected as suggested.

---

## Author Comment (AC2) · 19 Dec 2017

After reading the manuscript and the interactive comments by the other anonymous referee, I can say that my opinion about the paper submitted by Damian Insua-Costa and Gonzalo Miguez-Nacho is highly positive.

The origin of moisture to produce precipitation in a particular region is a very important meteorological problem. The authors of the present paper review some available methods of moisture origin assignation to the observed or modelled precipitation and/or precipitable water in their paper, and they propose a new method that they have incorporated to the WRF model. They validate the method through a month of integration (over US) and they apply it to analyse the interesting US Great Lakes snowstorm of November 2014. The paper is not only a good contribution to the main meteorological problem already mentioned, as well to the understanding of a very interesting particular case, but it is also a very well written paper, clear and with well-presented complementary figures.

We would like to thank very much the reviewer for his/her kind remarks and positive review. Please, find below a response to the specific comments.

I would accept the paper for publication almost as it is, although the comments of referee #1 can surely improve the text. I would only add a few small complementary details: Noting that the method is intrinsically coherent from the modelling point of view (the error of the addition of all the contributing origins into the total modelled precipitation is very small), it is worthy to compare the observed and modelled precipitation, in order to better evaluate the significance of the possible contribution of the different moisture origins to the observed/actual precipitation:

Fig. 12 does compare the observed and modelled total precipitation for the case of November 2014 in Great Lakes; why do not do it (in Fig. 5) to compare the observed and modelled precipitation (only on land, of course) during the whole validation month?

[Figure]

The figure above shows observed precipitation for the month of November 2014 from the CPC unified gauge-based analysis dataset of global daily precipitation. The model result in Fig 5a compares very favourably with the actual accumulations.

Xie, P., Chen, M., & Shi, W. (2010, January). CPC unified gauge-based analysis of global daily precipitation. In Preprints, 24th Conf. on Hydrology, Atlanta, GA, Amer. Meteor. Soc (Vol. 2).

We didn't show observed precipitation for the validation period because we do not want to confuse the reader with the concepts of validation for the method and for the WRF model simulation. What we validate first is the tracer method itself, regardless of whether WRF simulation results are realistic or not. We do not draw any conclusion about the origin of precipitation for the North American region during the month, and only discuss results in terms of the contribution of different sources to precipitation very briefly. We do test the method's ability to track moisture from different sources and to separate their contributions to precipitation.

Once we establish that the method is sound, then in the application example it is indeed very relevant to compare with observations, in order to verify that the simulation results are realistic. In the case study, the important conclusions are what can be said about the origin of precipitation, and not the method's ability to trace moisture, which was tested previously.

For these reasons, we would like to keep the clear separation between the validation of the method itself in an earlier section, where a comparison with precipitation observations is not so relevant, and a case study in a later section, where a validation of the model precipitation results is indeed very important.

Although it is clear in the text, perhaps in Fig. 13 it would be convenient to specify that the amount and percentage of precipitation represented in it is the part which origin is the Great Lakes evaporation.

The figure caption will be corrected.

Pg. 11, line 11: a mistake, 2104 (2014)

The typo will be corrected.

---

## Author Comment (AC3) · 19 Dec 2017

In general this is a very good paper, with an excellent background on the range of moisture tracking options available including Lagrangian and Eulerian tracking schemes, as well as a detailed explanation of the model, data sources, and the novel contributions of this team. The moisture tagging approach that is implemented within WRF is somewhat outside of my academic background, but in general appears sound. The validation of the approach against an observed extreme event is particularly interesting and it is a sign of how far the science has come in recent years, especially in realistically representing surface and atmosphere coupling during extremes.

We thank very much the reviewer for the positive review. Please, find below the response to the specific comments.

I have only a few minor comments:

1. First in the introduction, the authors briefly discuss offline Eulerian tracking schemes, and it is suggested that the vertical integration is a significant shortcoming of these approaches. I think that if the authors are going to cite the Goessling and Reick (2013) paper (which is critical of the single column version of the WAM-2layers, as described in Keys et al. 2012), the authors should also cite how this issue has been addressed using a two-layer, model-level version of the Eulerian tracking scheme, which performs favorably relative to regional climate model comparisons. One of the other reviewers already highlights this, by pointing the authors to van der Ent et al. (2013) "Should we use a simple or complex model for moisture recycling and atmospheric moisture tracking?" https://doi.org/10.5194/hess-17-4869-2013

Other work that has used the two-layer, model-level tracking scheme includes van der Ent et al. (2014) which couples the WAM-2layers to a land-surface hydrology model, Keys et al. (2014) which examines whether the WAM-2layers can be used with multiple datasets, and Duerinck et al. (2016) which examines soil moisture coupling in Illinois. I am by no means suggesting the authors cite this list of other papers, but rather am illustrating that much work has been done to address the single column assumption, and now in using the improved version.

I do recommend the authors consider adding a sentence or two more at Page 2, Line 17 to more accurately represent the current state of Eulerian tracking generally (and that as a 'class' of tracking schemes some Eulerian models have addressed the valid criticism associated with the single model level integration).

We totally agree with the reviewer and, therefore, the following sentence is going to be added on Page 2, Line 17: "However, in recent years this hypothesis has been relaxed by adding an additional vertical level to some offline Eulerian models (i.e., moving from a single column to two layers), which has considerably improved the results provided by this method (Van Der Ent et al., 2013)."

Van Der Ent, R. J., Tuinenburg, O. A., Knoche, H. R., Kunstmann, H., and Savenije, H. H.: Should we use a simple or complex model for moisture recycling and atmospheric moisture tracking?, Hydrology and Earth System Sciences, 17, 4869–4884,https://doi.org/10.5194/hess-17-4869-2013, 2013.

2. The authors make a point in the final sentence of the abstract by writing ". . .resulting in the highest socio-economic impacts." Since this is the final sentence in their abstract I think the authors ought to either:

a) explore this a bit more, clarifying what those socio-economic impacts actually were(in specific terms) during the snowstorm event, which populations were affected, and may be even the adequacy of alerts and warnings ahead of the snowstorm.

b) eliminate any reference to that aspect of the paper.

I think that the authors have done such an amazing job with the rest of this work that it seems a little bit like they are doing themselves and the reader a disservice by mentioning socio-economic impacts so blithely (aside from the mention at Page 15, 1stparagraph of section 4). I think it is the norm in this field to feel obligated to say something about socio-economic impacts since you have to justify why this science matters. At this point, if the justification is 'socio-economic impacts' then I'm not convinced that this science helps with anything. I think it could, such as through improved monitoring of lake temperatures, regional humidity, etc. and coupling such monitoring insights with emergency management and weather monitoring stations. Perhaps this was already done during the snowstorm. But I think that the authors ought to dig a bit deeper here, if they want to justify the paper as such.

Following the recommendations of the reviewer, we are going to add the following sentence on Page 16, Line 5: "which affected especially New York state (mainly cities bordering lakes Erie and Ontario, and in particular, the Buffalo area) between the 17th and 21st of this month, causing at least 13 fatalities, widespread food and gas shortages due to impassable roads and, in general, many other traffic problems and material losses derived from the storm (National Weather Service, NOAA, 2014)."

National Weather Service, NOAA, U. D. o. C.: Lake Effect Summary: November 17-19, 2014, https://www.weather.gov/buf/lake1415_stormb.html, 2014.

These events are usually well forecasted, perhaps with some uncertainty in the exact position of the snowbands originating from the lakes. We mentioned socioeconomic impacts not really thinking of improving alert systems or to justify our work, but to simply highlight the social relevance of the event. We chose this particular case study because lake effect snowstorms are an easy and clean example to illustrate how the tracer´s method works.

MINOR CORRECTIONS (Page = P, Line = L)

P1 L21 Change 'especial' to 'special'.

The typo will be corrected.

P3 l2 Check formatting for the citation.

The citation will be corrected.

P5 L7 The last clause of this sentence is confusing; consider revising for clarity.

The sentence ", some of the most commonly used and of contrasted performance in numerous situations. " is going to be replaced by "These schemes have been selected because they are some of the most commonly used and show a reliable performance in numerous situations."

P11 L11 Change '2104' to '2014'.

The typo will be corrected.

P11 L21 Change 'precipitations' to 'precipitation'

The typo will be corrected.

P15 L11 Good overview of the snowstorm event, but this is not adequate for justifying this work. Consider adding more substantive context for using this storm as a justification for the approach (perhaps in the summary section, or wherever is appropriate).

See response to general comment 2 above.

P16 L5 Cite the source of the "Snowvember" reference.

The colloquial nickname "Snowvember" spread quickly through the population and various local media after the event. There is no specific citation for it because the term does not have a specific known author.

P16 L17 Change 'Eire' to 'Erie'.

The typo will be corrected.